# PAX7 target genes are globally repressed in facioscapulohumeral muscular dystrophy skeletal muscle

Christopher R.S. Banerji[1,2,3,4], Maryna Panamarova[1], Husam Hebaishi[1], Robert B. White [1,5], Frédéric Relaix[6], Simone Severini[2,7] & Peter S. Zammit [1]

Facioscapulohumeral muscular dystrophy (FSHD) is a prevalent, incurable myopathy, linked to hypomethylation of *D4Z4* repeats on chromosome 4q causing expression of the DUX4 transcription factor. However, DUX4 is difficult to detect in FSHD muscle biopsies and it is debatable how robust changes in DUX4 target gene expression are as an FSHD biomarker. PAX7 is a master regulator of myogenesis that rescues DUX4-mediated apoptosis. Here, we show that suppression of PAX7 target genes is a hallmark of FSHD, and that it is as major a signature of FSHD muscle as DUX4 target gene expression. This is shown using meta-analysis of over six FSHD muscle biopsy gene expression studies, and validated by RNA-sequencing on FSHD patient-derived myoblasts. DUX4 also inhibits PAX7 from activating its transcriptional target genes and vice versa. Furthermore, PAX7 target gene repression can explain oxidative stress sensitivity and epigenetic changes in FSHD. Thus, PAX7 target gene repression is a hallmark of FSHD that should be considered in the investigation of FSHD pathology and therapy.

[1] Randall Centre of Cell and Molecular Biophysics, New Hunt's House, King's College London, Guy's Campus, London SE1 1UL, UK. [2] Department of Computer Science, University College London, London WC1E 6BT, UK. [3] Centre of Mathematics and Physics in the Life Sciences and Experimental Biology, University College London, London WC1E 6BT, UK. [4] Statistical Cancer Genomics, Paul O'Gorman Building, UCL Cancer Institute, University College London, London WC1E 6BT, UK. [5] School of Anatomy, Physiology & Human Biology, The University of Western Australia, Crawley, WA 6009, Australia. [6] Paris Est-Creteil University, IMRB U955, Faculté de médecine 8 rue du Général Sarrail, 94000 Créteil, France. [7] Institute of Natural Sciences, Shanghai Jiao Tong University, 800 Dongchuan Road, Minhang District, 200240 Shanghai, China. Correspondence and requests for materials should be addressed to C.R.S.B. (email: christopher.banerji15@imperial.ac.uk) or to P.S.Z. (email: peter.zammit@kcl.ac.uk)

Facioscapulohumeral muscular dystrophy (FSHD) is a prevalent inherited skeletal myopathy (12/100,000)[1] that typically presents during the second decade of life in males and third decade in females. FSHD is characterised by an asymmetric, descending skeletal muscle atrophy affecting specific muscle groups, including the orbicularis occuli/orbis, the scapular fixator muscles, the biceps and latterly, the tibialis anterior and certain other lower limb muscles[2]. Curiously, muscles such as the quadriceps, diaphragm and deltoids are spared[2, 3]. FSHD is also linked to retinal telangiectasia and sensorineural hearing loss, pointing to more systemic mechanisms[4, 5].

Genetically, FSHD is associated with hypomethylation of the D4Z4 macrosatellite repeat at chromosome 4q35, coupled with a permissive 4qA haplotype containing a polyadenylation signal in the pLAM region distal to the final D4Z4 repeat. Hypomethylation is caused either by D4Z4 repeat length truncation to between 1–10 D4Z4 units in FSHD1 (MIM 158900) (~ 95% of cases)[6] or by mutation in epigenetic modifiers in FSHD2 (MIM 158901), particularly *SMCHD1*[7], but with *DNMT3B* also recently identified in rare cases[8]. Each D4Z4 unit contains an open reading frame for a retrogene termed double homeobox 4 (*DUX4*) (MIM 606009). D4Z4 hypomethylation permits transcription of DUX4 from the final D4Z4 repeat, with *DUX4* message stabilised by addition of a polyadenylation tail by the signal in the flanking DNA on a permissive haplotype, and translated. The consensus is that such *DUX4* expression underlies pathology[9] and DUX4 target gene expression has been suggested as the major molecular signature in FSHD[10]. However, detection of DUX4 is difficult in FSHD muscle biopsies and a consistent molecular biomarker of FSHD muscle based on *DUX4* expression has yet to be fully validated.

DNA binding, and so target gene selection, is defined by the homeodomains of DUX4, which display significant amino-acid sequence homology to the homeodomains of the transcription factors PAX3 and PAX7[11]. Given this homology, it has been hypothesised that competitive inhibition of PAX3 and/or PAX7 by DUX4 may also contribute to FSHD pathology. PAX3 and PAX7 are critical to development in the early embryo[12], most notably playing essential, non-redundant roles, in the generation of the skeletal myogenic lineage[13], and derivatives of the dorsal neural ectoderm[12]. PAX3 and PAX7 are also highly homologous, with 86% sequence similarity[14] and are well conserved between mouse and man. In adult, *Pax7* is expressed in quiescent, activated and proliferating muscle satellite cells[15], while the *Pax3* locus is active in some satellite cells in certain muscles, such as the diaphragm, and may be transiently expressed during activation[13, 16–18]. Both genes promote cell survival and proliferation, while preventing precocious differentiation[15, 19]. In man, PAX7 is also detected in satellite cells[20, 21] but little is known about PAX3 in postnatal skeletal muscle. However, mutations in *PAX3* in man are associated with Waardenburg syndrome[22], a condition whose characteristics include high frequency, sensorineural hearing loss, a symptom also reported in FSHD.

In line with a competitive inhibition model, it has been demonstrated that cytotoxicity caused by expression of DUX4 or its murine ortholog mDUX is rescued by over-expression of Pax3 or Pax7 in murine myoblasts[11, 23]. We have recently shown in a transgenic mouse model of FSHD that DUX4 levels rise during the early phases of muscle regeneration in activated and proliferating satellite cells, a phase when Pax7 is operating[24]. It has also been shown that FSHD patient-derived cells express high levels of DUX4 during a muscle progenitor phase characterised by high levels of PAX3 and PAX7 during in vitro differentiation of embryonic stem (ES) cells, providing another time point in FSHD myogenic differentiation where competitive inhibition may occur[25]. A more recent study of myogenic differentiation of ES

cells and induced pluripotent stem (iPS) cells from FSHD patients showed that although PAX7 and DUX4 mRNA are found in a myogenic progenitor phase, the two proteins are not present in the same cell[26]. Interestingly, the frequency of co-expression of PAX7 and DUX4 in FSHD cells was significantly lower than expected, indicating that some non-competitive inhibitory interaction between the two proteins may ensure such mutual exclusivity[26]. PAX7 also drives long term epigenetic changes associated with de-repression of gene expression, including DNA demethylation[27]. Hence, inhibition of PAX7 by DUX4 in FSHD during a muscle progenitor stage may result in a global suppression of PAX7 transcriptional targets that persists in terminally differentiated FSHD muscle. Such repression may represent a novel biomarker of FSHD muscle, and define a new set of therapeutic target genes. Examination of PAX7 target gene expression in FSHD patient-derived samples is lacking, however, and so the translational implications of this mechanism are unresolved.

Here, we adopt a meta-analysis approach across six independent FSHD muscle biopsy studies to analyse several FSHD molecular biomarkers based on DUX4 target gene expression, and to compare them to a novel biomarker based on Pax7 target gene repression. We find that only the new biomarker based on Pax7 target gene repression is consistently able to discriminate FSHD from control skeletal muscle biopsies in every FSHD biopsy data set, while the discriminatory power of DUX4 target gene expression is limited to just two studies[10, 28]. The discriminatory power of Pax7 target gene repression is verified via RNA-seq of immortalised human myoblasts isolated from FSHD patients and matched controls. We also show that the Pax7 target genes repressed in FSHD are enriched for factors that suppress the HIF1α-mediated hypoxic response, indicating over-activation of HIF1α in FSHD as a putative pathomechanism driving oxidative stress sensitivity, as we previously suggested[29]. Lastly, co-expression of DUX4 and Pax7 results in repression of their respective target genes, confirming mutual target gene inhibition between these two transcription factors. Overall, our findings demonstrate that Pax7 target gene repression is at least as strong a signature in FSHD skeletal muscle as DUX4 target gene over-expression. Thus, Pax7 target genes should be considered in the investigation of pathology and design of therapies in FSHD.

## Results

**DUX4 target gene expression as an FSHD biomarker.** Transcriptomic biomarkers of FSHD muscle are important both for understanding molecular processes perturbed in pathology and for monitoring therapeutic responses in clinical trials[3]. In recent years, a number of transcriptomic biomarkers for identifying FSHD muscle biopsies have been proposed, notably the biopsy-derived 15 gene signature described by Rahimov et al.[3], and the 114 gene signature derived from human myoblasts over-expressing DUX4, described by Yao et al.[10]. Outside of a small number of samples, however, these biomarkers have received little validation.

We identified six independent transcriptomic studies profiling FSHD muscle biopsies alongside suitable controls (five microarray studies totalling 82 FSHD and 82 controls and one RNA-seq study with 15 FSHD and 8 control samples)[3, 10, 28, 30–32] and assessed the discriminatory capacity of the Rahimov et al.[3] and Yao et al.[10] biomarkers via meta-analysis (Fig. 1). Meta-analysis over five independent data sets revealed that neither biomarker could reliably discriminate FSHD from control samples. Interestingly however, both signatures were able to discriminate between FSHD and control muscle biopsies on one microarray data set independent of their discovery data sets, namely that described by Tasca et al.[28]. Unlike the other studies, this data set

profiled affected FSHD muscle as confirmed by magnetic resonance imaging (MRI), indicating that an MRI guided muscle biopsy approach may be optimal for identifying tissue displaying molecular changes associated with FSHD.

The DUX4-based biomarker of Yao et al.[10] was derived from RNA-seq data of two human myoblast samples over-expressing DUX4, compared to two qualitatively different controls, one of which was contaminated and contained reads from a DUX4-expressing sample[33]. Hence it is unclear how representative the Yao et al.[10] biomarker is of DUX4 expression. Unfortunately, the only data set profiling FSHD muscle biopsies by RNA-seq was also produced by Yao et al.[10] and our other validation data sets were profiled by microarray. Therefore, it is also unclear whether failure of these genes to discriminate FSHD status is due to DUX4 target genes not being a biomarker, or due to differences in the technology platforms used, namely microarray vs. RNA-seq.

To address these issues, we derived two further DUX4 over-expression signatures from two independent human myoblast DUX4 over-expression studies. The first signature was derived

from a recent RNA-seq data set profiling human myoblasts over-expressing DUX4 for 8 h via a doxycycline inducible promotor against suitable controls, as described by Choi et al.[34] We identified 212 transcripts that were significantly upregulated in DUX4-expressing samples (FDR < 0.05, log FC > 2, Supplementary Data 1). The Choi et al.[34] DUX4 target gene signature was significantly elevated in FSHD samples from the Yao et al.[10] RNA-seq muscle biopsy data set. However, we found no significant difference in Choi et al.[34] DUX4 target levels between FSHD and control samples in any of the five microarray data sets or on meta-analysis (Fig. 2a–b).

A second DUX4 signature was derived from a microarray data set published by Geng et al.[35], profiling DUX4 lentivirus transduced human myoblasts after 24 h against suitable controls[35]. We identified 165 transcripts that were significantly upregulated in DUX4-expressing samples (FDR < 0.05, log FC > 2, Supplementary Data 2). The Geng et al.[35] DUX4 target gene signature was significantly upregulated on the FSHD muscle biopsy samples from the Yao et al.[10] RNA-seq data set.

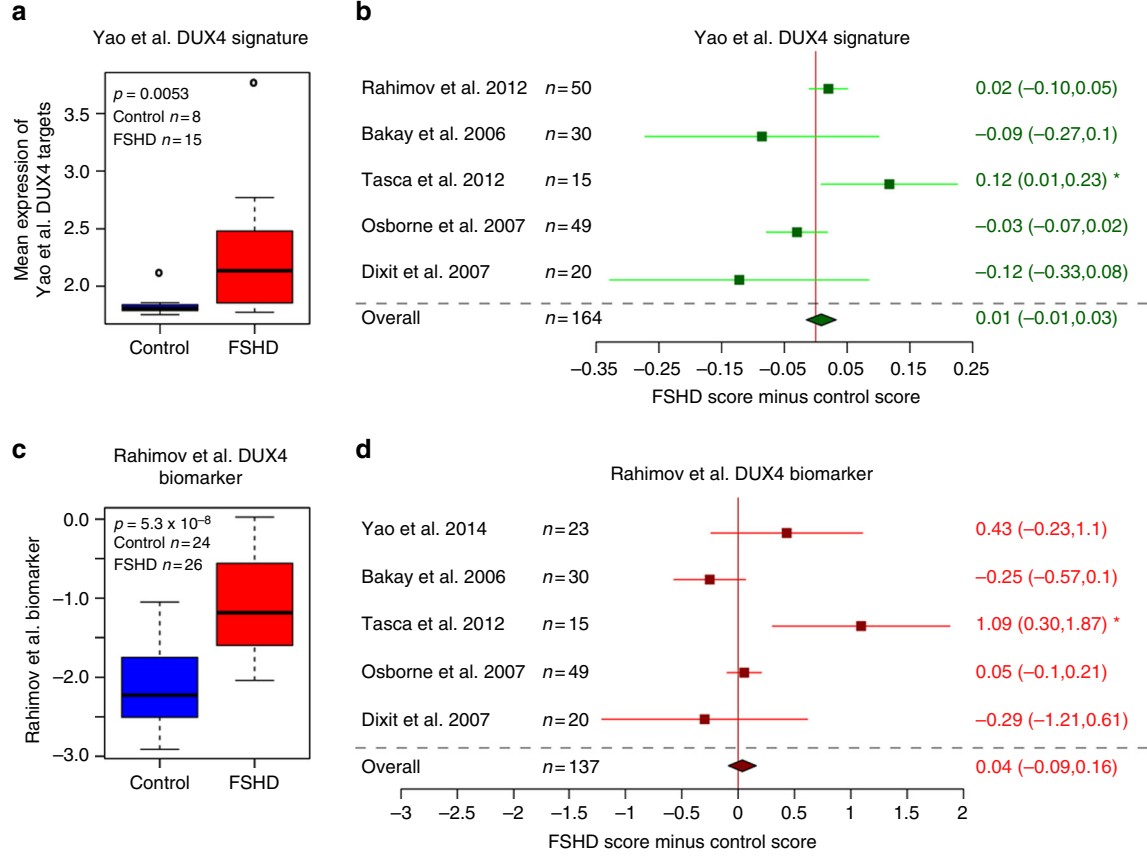

**Fig. 1** Reported FSHD biomarkers do not validate on most published FSHD muscle biopsy gene expression data sets. **a** Box plot confirms that the Yao et al.[10] DUX4 114 target gene signature validates as a biomarker on the RNA-seq FSHD muscle biopsy data set published by Yao et al.[10]. The box represents the interquartile range (IQR), with the median indicated by a line. Whiskers denote min (1.5*IQR, max (observed value)). ''o'' represents data points greater than 1.5 IQR from the median, n = 15 FSHD and n = 8 control muscle biopsies. **b** A forest plot displays the results of meta-analysis of the discriminatory power of the Yao et al.[10] DUX4 target gene signature across five published microarray FSHD muscle biopsy data sets (in total n = 82 FSHD and n = 82 control muscle biopsies). The differential scores (FSHD score minus control score) alongside 95% confidence intervals are provided. **c** A box plot confirms that the Rahimov et al.[3] FSHD 15 gene biomarker validates as a biomarker on the microarray FSHD muscle biopsy discovery data set published by Rahimov et al.[3]. The box represents the interquartile range (IQR), with the median indicated by a line. Whiskers denote min (1.5*IQR, max (observed value)). ''o'' represents data points greater than 1.5 IQR from the median, n = 26 FSHD and n = 24 control muscle biopsies. **d** A forest plot displays the results of meta-analysis of the discriminatory power of the Rahimov et al.[3] FSHD biomarker across four published microarray and one RNA-seq FSHD muscle biopsy data sets (in total n = 71 FSHD and n = 66 control muscle biopsies). The differential scores (FSHD score minus control score) alongside 95% confidence intervals are provided. Neither biomarker is able to discriminate FSHD from control samples on meta-analysis. However, both are able to discriminate on the Tasca et al.[28] MRI-guided muscle biopsy microarray data set. For single studies a two-tailed Wilcoxon U-test was performed to assess significance, while a Fisher's combined test was employed for overall assessment: either the p-value is given, or an asterisk denotes p < 0.05

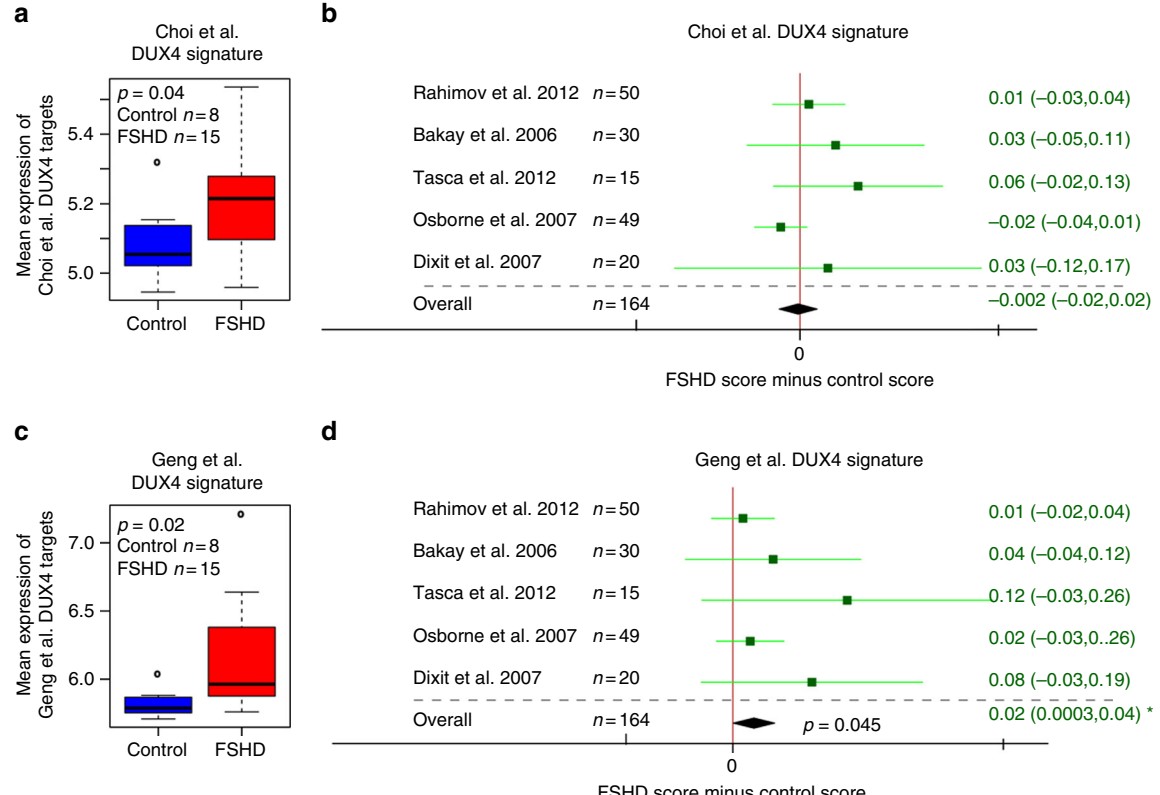

**Fig. 2** Novel DUX4 target gene signatures only discriminate some FSHD muscle biopsy gene expression data sets from controls. **a** A box plot demonstrates that the Choi et al.[34] RNA-seq-based 212 DUX4 target gene signature validates as a biomarker on the RNA-seq FSHD muscle biopsy data set published by Yao et al.[10]. The box represents the interquartile range (IQR), with the median indicated by a line. Whiskers denote min (1.5*IQR, max (observed value)). "o" represents data points greater than 1.5 IQR from the median, $n = 15$ FSHD and $n = 8$ control muscle biopsies. **b** A forest plot displays the results of meta-analysis of the discriminatory power of the Choi et al.[34] DUX4 target gene signature across five published microarray FSHD muscle biopsy data sets (in total $n = 82$ FSHD and $n = 82$ control muscle biopsies). The differential scores (FSHD score minus control score) alongside 95% confidence intervals are provided. The Choi et al.[34] DUX4 target gene signature is not a significant biomarker on any microarray data set nor on meta-analysis. Note: in the Choi et al.[34] study, target genes were analysed after 8 h of DUX4 induction. **c** A box plot demonstrates that the Geng et al.[35] microarray based 165 DUX4 target gene signature validates as a biomarker on the RNA-seq FSHD muscle biopsy data set published by Yao et al.[10]. The box represents the interquartile range (IQR), with the median indicated by a line. Whiskers denote min (1.5*IQR, max (observed value)). "o" represents data points greater than 1.5 IQR from the median, $n = 15$ FSHD and $n = 8$ control muscle biopsies. **d** A forest plot displays the results of meta-analysis of the discriminatory power of the Geng et al.[35] DUX4 target gene signature across five published microarray FSHD muscle biopsy data sets (in total $n = 82$ FSHD and $n = 82$ control muscle biopsies). The differential scores (FSHD score minus control score) alongside 95% confidence intervals are provided. The Geng et al.[35] DUX4 target gene signature is not a significant biomarker on any individual microarray data set, however, it is significant on meta-analysis. For single studies a two-tailed Wilcoxon U-test was performed to assess significance, while a Fisher's combined test was employed for overall assessment: either the p-value is given, or an asterisk denotes $p < 0.05$

Moreover, across the five microarray FSHD studies, although no individual study showed statistically significant elevation of DUX4 target gene expression, all studies showed a positive trend towards this. On meta-analysis, we identified a small, but significant, upregulation of DUX4 targets in FSHD samples (Fisher's combined test $p = 0.045$, Fig. 2c–d).

As DUX4 suppresses MyoD[11, 24] we also evaluated expression of MyoD target genes identified by de la Serna et al.[36] in the five FSHD microarray studies. Although no individual data set was significant, there was a trend towards repression of MyoD target genes across microarray studies of FSHD muscle biopsies, but this was also not significant on meta-analysis (Supplementary Fig. 1)

Our results indicate that DUX4 target gene expression is a weak, but significant, biomarker of FSHD status but points to a lack of compatibility between RNA-seq and microarray studies for evaluating DUX4 transcriptional target genes. Despite overlap between the three DUX4 signatures (28 genes shared between Geng et al.[35] and Choi et al.[34], 45 between Yao et al.[10] and Geng et al.[35] and 29 between Choi et al.[34] and Yao et al.[10]), we see that

RNA-seq-derived DUX4 upregulated target genes are only capable of discriminating FSHD biopsies from controls, if the biopsies were also profiled by RNA-seq. In contrast, microarray-based DUX4 upregulated target genes show discriminatory power on both microarray (on meta-analysis) and RNA-seq data sets (Figs. 1 and 2). This is likely due to differences in coverage and dynamic range across the two technologies and is a consideration when evaluating DUX4 transcriptional target genes on FSHD samples.

**PAX7 target gene repression hallmarks FSHD skeletal muscle.** We next considered repression of PAX7 target genes as a biomarker of FSHD muscle. To derive a set of PAX7 target genes, we assayed primary murine satellite cell-derived myoblasts overexpressing Pax7, a dominant negative Pax7 fusion protein, comprising the DNA binding domains of Pax7 and the engrailed repressor domain (Pax7-ERD)[15] and a control retroviral construct. PAX7 is highly conserved between mouse and man with 97% amino-acid sequence homology[13], hence a conserved set of

target genes is expected. We assayed gene expression via micro-array to facilitate compatibility with the publically available FSHD muscle biopsy data sets. Microarray data were pre-processed and normalised as described in Methods. Hierarchical clustering and principal component analysis confirmed reproducibility of the transcriptional landscapes induced by the PAX7 constructs and close clustering of replicates.

Unlike DUX4, which is a potent transcriptional activator[24, 29], PAX7 likely modulates transcription in more complex ways[12]. Hence, to derive a biomarker of *Pax7* expression in murine satellite cells, rather than focus only on the most strongly induced target genes, we performed differential expression analysis to derive a set of 311 upregulated target genes (defined as induced by PAX7 over-expression and suppressed by PAX7-ERD) and a set of 290 downregulated target genes (defined as suppressed by PAX7 over-expression and induced by PAX7-ERD) (Supplementary Data 3). A biomarker of PAX7 was then defined from consideration of the ratio of mean upregulated to downregulated target gene expression in a given sample (see Methods). To validate this ratio as a biomarker of PAX7, we evaluated it on an independent microarray dataset describing *Pax7* retroviral over-expression in primary murine satellite cells alongside control retrovirus in triplicate[14]. Our PAX7 biomarker demonstrated significantly higher values on samples over-expressing *Pax7* in this independent data set, so confirming its validity (Supplementary Fig. 2).

To evaluate the functional relevance of PAX7 target genes to FSHD pathology, we performed gene set enrichment analysis (GSEA) of activated and repressed target genes of PAX7 separately against the Molecular Signatures database (MSigDB)[37, 38]. The top 100 enriched gene sets for the PAX7 targets are presented in Supplementary Data 4 and 5. To refine our search we identified gene sets which displayed inverse enrichments across activated and repressed targets (Fig. 3a). PAX7 is associated with repression of a hypoxia gene set. Importantly of 48 hypoxia gene sets in MSigDB[37, 38], only one was identified, describing genes upregulated or downregulated by over-expression of an active form of HIF1α[39]. We have previously implicated over-activation of HIF1α as a putative mechanism for oxidative stress sensitivity in FSHD[29]. Another gene set repressed by PAX7 is EZH2 target genes (Fig. 3a). EZH2 is a critical component of the polycomb repressor complex 2 (PRC2), which has been associated with perturbed methylation in FSHD[40, 41]. Thus PAX7 target genes are enriched for pathways previously identified in FSHD molecular pathology.

We next evaluated our PAX7 biomarker on the six FSHD muscle biopsy data sets. We found that the levels of PAX7 target genes were significantly repressed in FSHD muscle biopsy samples profiled by RNA-seq (Fig. 3b). Importantly, PAX7 target gene repression was also found to be a significant biomarker of FSHD status in each of the five microarray FSHD muscle biopsy studies independently, leading to a highly significant repression of PAX7 target genes on meta-analysis (Fisher's combined test $p = 3.5 \times 10^{-9}$, Fig. 3c).

To confirm that PAX7 target gene repression is specific to FSHD and not attributable to general muscle wasting, we identified four published microarray studies profiling Duchenne muscular dystrophy (DMD) muscle biopsies alongside matched controls[31, 42, 43]. Meta-analysis revealed no consistent significant difference in PAX7 target gene expression between DMD muscle and controls (Supplementary Fig. 3).

The PAX7 and DUX4 biomarkers were derived differently, so it is necessary to consider how the results would change if this were not the case. Firstly, if repressed DUX4 targets are combined with the induced targets to construct a biomarker of DUX4 analogous to our derivation of the PAX7 biomarker, DUX4 target genes are lost as a discriminator of FSHD status. This indicates that only robustly upregulated DUX4 targets have discriminatory power in FSHD and that suppressed targets introduce noise that masks this signal. Conversely, if we separately consider the upregulated or downregulated target genes of PAX7, rather than together as a biomarker of PAX7 (analogous to the DUX4 biomarker), we see that upregulated target genes are significantly repressed in FSHD microarray samples on meta-analysis, while downregulated target genes are significantly over-expressed, confirming that both activated and suppressed PAX7 target genes are perturbed in FSHD. However, the upregulated or down-regulated target genes alone are unable to discriminate between FSHD and control muscle biopsies on the RNA-seq data set (Supplementary Fig. 4).

This cross comparison demonstrates that the biomarker construction which maximises DUX4 target gene discriminatory power in FSHD focuses only on upregulated targets, while that which maximises PAX7 target gene discriminatory power utilises both upregulated and downregulated target genes.

To confirm that our PAX7 biomarker does not discriminate FSHD status from control by chance, we performed a resampling procedure, selecting 1000 random gene sets of equivalent size to both the upregulated and downregulated PAX7 target gene sets. For each randomisation, we defined a biomarker analogously to the PAX7 biomarker using the ratio of upregulated to down-regulated target genes and evaluated its capacity to discriminate FSHD and control muscle biopsies on meta-analysis across the five FSHD microarray muscle biopsy data sets. None of the random biomarkers were able to significantly discriminate FSHD from control samples on meta-analysis (Supplementary Fig. 5), hence the probability that PAX7 target gene repression in FSHD is attributed to chance is less than 1/1000.

**PAX7 or DUX4 target genes form equivalent FSHD bio-markers on RNA-seq data.** To compare the discriminatory capacity of our PAX7 target gene biomarker against the various DUX4 target gene biomarkers, we employed a receiver operating characteristic (ROC) approach. Given the data compatibility issues explored above, we considered the microarray and RNA-seq muscle biopsy data separately. An impact of this is that analysis over the RNA-seq data is underpowered relative to the microarray data due to large differences in sample size (RNA-seq data: 8 controls, 15 FSHD; pooled microarray data: 82 controls, 82 FSHD), and such a discrepancy should be taken into account on data interpretation.

Considering the microarray data, the only significant biomarkers on meta-analysis were our PAX7 target gene repression signature and the robustly upregulated DUX4 targets described by Geng et al.[35] while Choi et al.[34] and Yao et al.[10] failed to reach significance. Both biomarkers were computed for each sample on each microarray study independently and *z*-normalised within each study to ensure identically distributed predictors, studies were then pooled for ROC analysis. We found that PAX7 target gene repression significantly outperformed DUX4 target expression as a classifier of FSHD status (Pax7 AUC = 0.81, DUX4 AUC = 0.59, DeLong's test: $p < 5.7 \times 10^{-6}$, Fig. 4a), indicating that in microarray data, PAX7 target gene repression is a more robust signature of FSHD skeletal muscle than DUX4 target gene expression.

Considering the smaller RNA-seq data set, all three DUX4 upregulated target gene signatures (Geng et al.[35], Choi et al.[34] and Yao et al.[10]), as well as our PAX7 target gene repression signature, proved to be significant biomarkers of FSHD status. Though the Yao et al.[10] signature demonstrated a slightly higher AUC on the RNA-seq samples, than the Pax7 biomarker (Yao et al.[10]

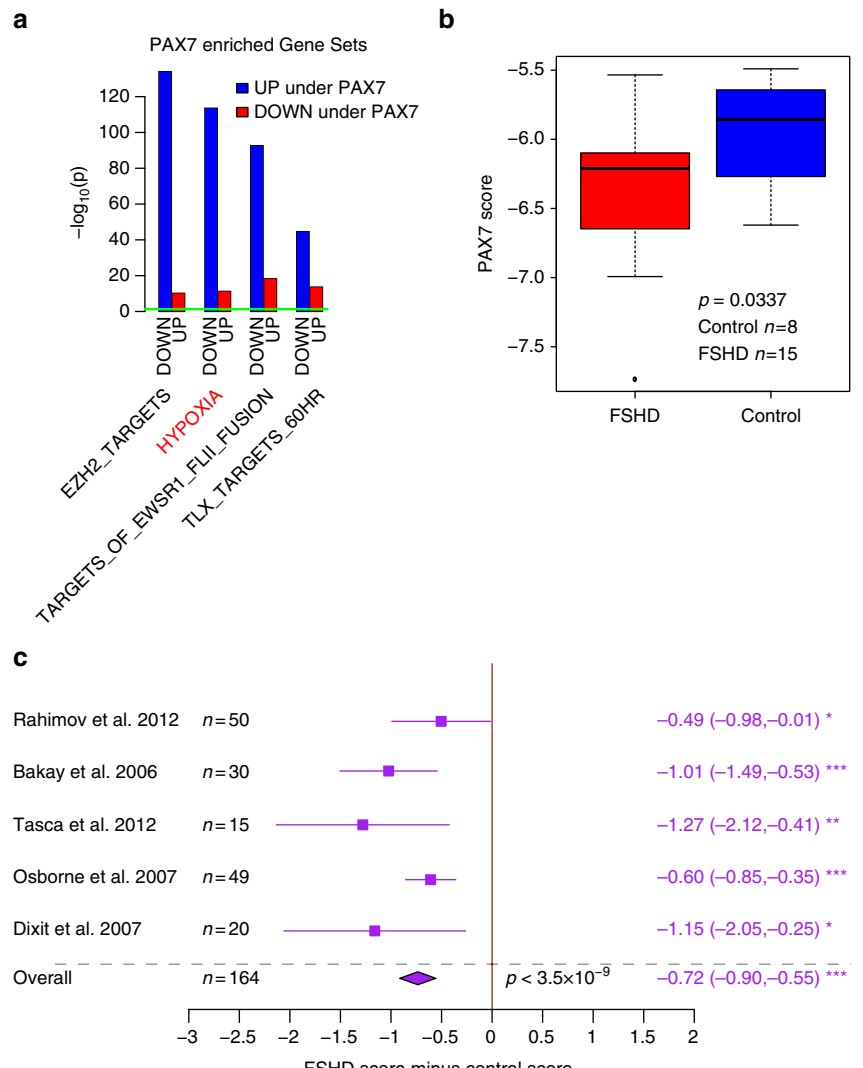

**Fig. 3** PAX7 target gene signature is a robust hallmark of FSHD skeletal muscle. **a** Gene set enrichment analysis was performed separately for activated and repressed targets of PAX7 against the Molecular Signatures database (MSigDB)[37, 38]. Gene sets in MSigDB are frequently partitioned into "genes upregulated in phenotype" and "genes downregulated in phenotype". If such opposing gene sets were inversely significantly enriched across activated and repressed PAX7 targets (as assessed by Fisher's exact test), they were considered robust. A bar plot displays the $-\log_{10}$ (enrichment p-value) for PAX7 robust gene sets. A horizontal green line denotes $p = 0.05$. UP and DOWN on the x-axis refers to the partition of labeled gene sets into genes positively and negatively associated with the gene set respectively. PAX7 targets are enriched for repression of a HYPOXIA gene set, which specifically describes genes upregulated and downregulated following over-expression of a constitutively active form of HIF1α. Another gene set affected by PAX7 repression is EZH2 target genes. **b** A box plot demonstrates that the PAX7 target gene signature derived from 311 upregulated target genes and 290 downregulated target genes, validates as a biomarker on the RNA-seq FSHD muscle biopsy data set published by Yao et al.[10]. The box represents the interquartile range (IQR), with the median indicated by a line. Whiskers denote min (1.5*IQR, max (observed value)). "o" represents data points greater than 1.5 IQR from the median, $n = 15$ FSHD and $n = 8$ control muscle biopsies. The two-tailed Wilcoxon U-test p-value is given. **c** A forest plot displays the results of meta-analysis of the discriminatory power of the PAX7 target gene signature across five published microarray FSHD muscle biopsy data sets (in total $n = 82$ FSHD and $n = 82$ control muscle biopsies). The differential scores (FSHD score minus control score) alongside 95% confidence intervals are provided. Our PAX7 target gene signature is a significant biomarker on every individual FSHD muscle biopsy data set, and is strongly significant on meta-analysis. For single studies, a two-tailed Wilcoxon U-test was performed to assess significance, while a Fisher's combined test was employed for overall assessment: *denotes $p < 0.05$, **denotes $p < 0.01$ and ***denotes $p < 0.001$

AUC = 0.85, Geng et al.[35] AUC = 0.79, Choi et al.[34] AUC = 0.78, PAX7 AUC = 0.78, Fig. 4b), DeLong's test revealed no significant differences between the discriminatory power of any of the biomarkers on this data set. Thus PAX7 target gene repression is at least as major a signature of FSHD skeletal muscle as DUX4 target gene expression.

**PAX7 or DUX4 target genes are equivalent biomarkers on FSHD myoblasts.** As muscle biopsies do not represent a pure

myogenic population of cells, we next investigated our PAX7 or DUX4 target gene biomarkers in FSHD patient-derived myoblast cell lines. We performed RNA-sequencing on immortalised human myoblasts isolated from three independent FSHD patients alongside matched controls[44, 45]. One of the FSHD patients considered is mosaic for the FSHD genotype and multiple clones are available[45], of which we considered five. These clones are isogenic with exception of the D4Z4 region, which is truncated to three repeats in clones 54–12, 54-2 and 54-A5 (an FSHD genotype), whereas clones 54-6

 

**a**

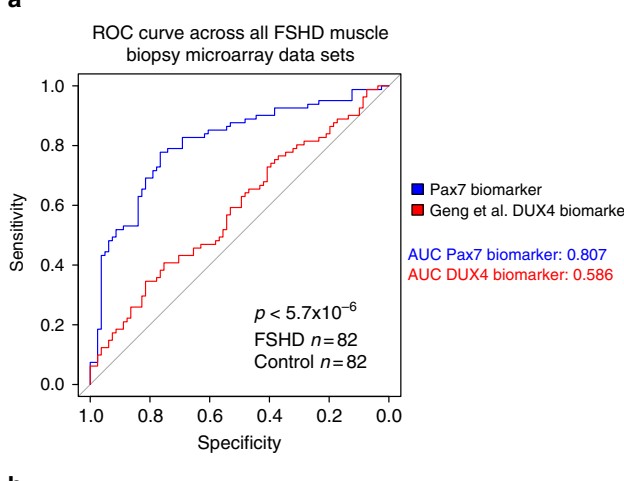

**b**

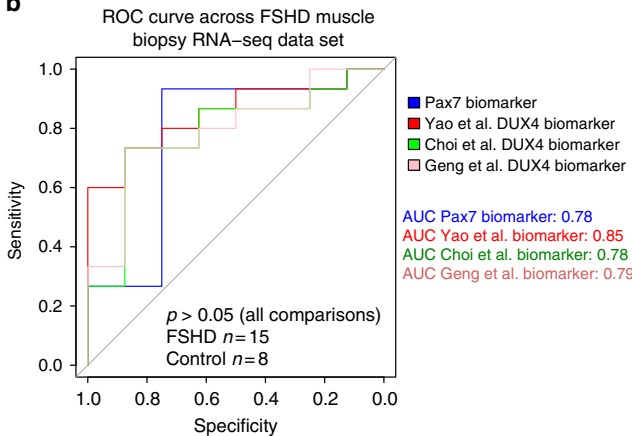

**Fig. 4** PAX7 target gene repression is an equivalent FSHD biomarker to DUX4 target gene expression. **a** A ROC curve compares the discriminatory power of our PAX7 biomarker with the DUX4 target gene signature that we derived from Geng et al.[35], across five microarray data sets (in total $n = 82$ FSHD and $n = 82$ control muscle biopsies). De-Long's test $p$-value is given and demonstrates that the PAX7 biomarker is a significantly better discriminator of FSHD status. **b** A ROC curve compares the discriminatory power of our PAX7 biomarker with the DUX4 target gene signatures that we derived from Geng et al.[35] and Choi et al.[34] and that described in Yao et al.[10] across the RNA-seq data set of $n = 15$ FSHD and $n = 8$ control muscle biopsies. De-Long's test reveals no significant differences in the discriminatory power of these four biomarkers

and 54-A10 have 13 repeats (a "healthy" number). The two further FSHD patient cell lines considered were the 12Abic and 16Abic immortalised myoblasts obtained from the UMMS Wellstone centre for FSHD, Worcester, USA, alongside sibling-matched controls[44]. RNA-sequencing was performed in triplicate on confluent myoblasts for each cell line, giving a total of 15 FSHD samples (three patients corresponding to five lines in triplicate) and 12 control samples (three individuals corresponding to four lines in triplicate).

Our PAX7 biomarker, alongside the DUX4 target gene signatures (Yao et al.[10], Choi et al.[34] and Geng et al.[35]), were computed for each sample and scores for each patient and matched control were $z$-normalised and pooled for analysis. We found that our PAX7 biomarker, the Yao et al.[10] and Geng et al.[35] DUX4 target gene signatures were all significant discriminators of FSHD status (Wilcoxon signed rank test $p < 0.05$, Fig. 5a–d), in line with our results from the muscle biopsy RNA-seq data, but the Choi et al.[34] DUX4 target gene signature was not (Fig. 5a). ROC curve analysis demonstrated that there was no significant

difference between the discriminatory power of our PAX7 biomarker and the DUX4 biomarkers on FSHD patient cell lines (De-Long's test $p > 0.05$, Fig. 5e). This provides further evidence that PAX7 target gene repression is at least as major a signature as DUX4 target gene expression in FSHD skeletal muscle.

**PAX7 and DUX4 mutually inhibit target gene activation.** Having established that PAX7 target genes are repressed in FSHD skeletal muscle, we next investigated if DUX4 could lead to repression of PAX7 transcriptional target genes, as suggested by the homology between their homeodomains[11]. A reporter construct assay was performed on human embryonic kidney (HEK-293) or NIH-3T3 cells transfected with plasmids encoding *DUX4*, *Pax7*, both *DUX4* and *Pax7* together, dominant negative *DUX4* (*DUX4-ERD*)[29] or PAX7 (*Pax7-ERD*)[15], and *GFP* as a control. To measure the ability of PAX7 to activate its transcriptional targets, we measured activity of the p34 reporter construct, consisting of concatermerised PAX3/7-binding sites driving *thymidine kinase-nlacZ*[15], along with endogenous expression of the PAX7 target gene *SELP*[14]. To quantify DUX4 activity, we used DUX4 reporter constructs *RFPL4B-luc* or *ZSCAN4-luc* controlling a luciferase reporter gene[46] and also measured endogenous expression of the DUX4 target genes *ZSCAN4*, *RFPL4B*, *MBD3L2* and *TRIM48*.

PAX7 robustly elevated p34 reporter activity, as expected. Interestingly, *DUX4* also increased p34 reporter activity, but to a smaller extent than PAX7. Strikingly both DUX4 and PAX7 proteins together resulted in no change of p34 reporter activity, with the level unaltered from that achieved by transfection of control GFP-encoding plasmid (Fig. 6a). Reverse transcription quantitative PCR (RT-qPCR) of PAX7 target gene *SELP* also confirmed its activation by PAX7, but when both DUX4 and PAX7 were present together, transcription of *SELP* dropped to control levels (Fig. 6b). Thus, while DUX4 can bind and activate the PAX7 reporter gene, when DUX4 is present with PAX7, the effect is to repress activation of PAX7 transcriptional target genes. Levels of PAX7 target gene activation achieved with DUX4 and PAX7 co-expression are lower than achieved by either PAX7 or DUX4 alone, implying that the mechanism by which DUX4 represses PAX7 transcriptional target gene activation may encompass mechanisms in addition to competitive inhibition of DNA-binding.

Performing reciprocal experiments using DUX4 reporters, we found that while DUX4 activated both the *RFPL4B-luc* or *ZSCAN4-luc* reporters, *PAX7* did not activate either DUX4-reporter (Fig. 6c, d). As observed with the PAX7 reporter assay, expression of both PAX7 and DUX4 proteins together, significantly reduced DUX4 reporter activity compared to DUX4 alone, and back to control levels with the *RFPL4B-luc* reporter (Fig. 6d). These findings were confirmed via RT-qPCR of endogenous DUX4 transcriptional target genes, which showed that DUX4 induced expression of *ZSCAN4*, *RFPL4B*, *MBD3L2* and *TRIM48*, while both DUX4 and PAX7 together failed to increase transcript levels of these DUX4 target genes to the same level as DUX4 alone, and in 3 out of 4 cases, not above control levels (Fig. 6e–h).

Therefore, inhibition between DUX4 and PAX7 proteins is reciprocal, whereby co-expression of PAX7 and DUX4 acts to suppress the respective transcriptional target genes of both DUX4 and PAX7.

## Discussion

Here, we demonstrate that PAX7 target gene repression is a hallmark of FSHD skeletal muscle. Moreover, it is a superior FSHD biomarker to DUX4 target gene expression across microarray data sets, and at least as major a signature of FSHD skeletal muscle when considering RNA-seq data.

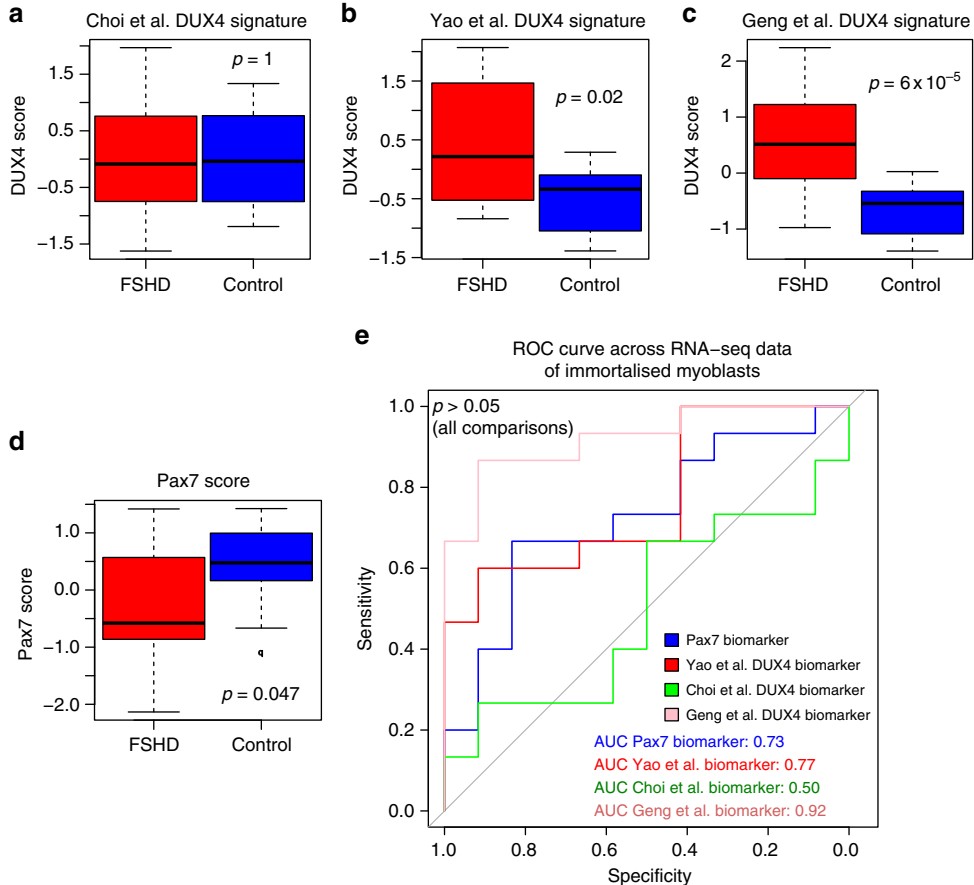

**Fig. 5** PAX7 and DUX4 target gene biomarkers validate on FSHD myoblast gene expression data sets. Box plots show the **a** Choi et al.[34], **b** Yao et al.[10] and **c** Geng et al.[35] DUX4 target gene signatures and **d** our PAX7 biomarker, computed for five FSHD patient lines and four matched control lines corresponding to three patients and controls RNA-sequenced in triplicate (in total $n = 15$ FSHD and $n = 12$ Control). In line with previous findings, PAX7 repression and DUX4 expression as assessed by the Yao et al.[10] and Geng et al.[35] DUX4 target signatures are significant biomarkers of FSHD status. The Choi et al.[34] DUX4 target gene biomarker is not a significant discriminator of FSHD status, where target genes were identified after 8 h of DUX4 induction. The box represents the interquartile range (IQR), with the median indicated by a line. Whiskers denote min (1.5*IQR, max (observed value)). "o" represents data points greater than 1.5 IQR from the median. The two-tailed Wilcoxon U-test p-value is given. **e** A ROC curve compares the discriminatory power of our PAX7 biomarker with the Geng et al.[35], Choi et al.[34] and Yao et al.[10] DUX4 target gene signatures across the RNA-seq data set of FSHD and control immortalised myoblasts. De-Long's test reveals no significant differences in the discriminatory power of these three DUX4 and one PAX7 biomarkers

DUX4 is currently the leading candidate gene in FSHD molecular pathology[9]. Recently, it has been suggested that DUX4 target gene expression is also the major molecular signature in FSHD skeletal muscle[43]. This has led to an acceleration of investigations into DUX4 target genes in FSHD, which are pro-apoptotic and suppress myogenic progression[11, 24, 29, 47–49]. In addition to these DUX4 focused studies, others have compared FSHD and control skeletal muscle in an hypothesis free manner. Using this approach, we and others, have revealed FSHD molecular mechanisms, which may not be direct consequences of DUX4 target gene expression, such as HIF1α target gene activation[29, 50] and epigenetic alterations in PCR2 genes[41]. Such mechanisms are an important line of investigation into FSHD pathology and therapeutics, but may not be fully understood by investigation of DUX4 target genes alone.

The idea that DUX4 and PAX7 protein may mutually inhibit the activation of their respective transcriptional target genes was originally proposed by Bosnakovski et al.[11] and was based on sequence similarity between the homeodomains of the two transcription factors. This model provides an alternative mechanism by which DUX4 may drive molecular changes in FSHD. However, besides the demonstration that Pax7 can

mitigate DUX4 driven apoptosis in murine myoblasts[11, 23], the role of PAX7 in human FSHD pathology has been little investigated. Recently, it has been demonstrated by several groups, including our own, that *DUX4* and *PAX7* mRNA are co-expressed in muscle progenitor cells from FSHD patients[25] and during regeneration in an FSHD mouse model[24], providing opportunity for the proteins to interact. A recent study investigating myogenic differentiation of ES and iPS cells derived from FSHD and control patients, reported that both *PAX7* and *DUX4* mRNA expression could be detected both during a myogenic precursor phase and in myotubes[26]. However, immuno-fluorescence suggested that PAX7 and DUX4 protein were not localised to the same cells. Although embryonic muscle is usually unaffected in FSHD[9], it is interesting that the authors could not find cells containing both PAX7 and DUX4. The statistically significant low frequency of co-expression indicates that PAX7 and DUX4 protein may interact to cause mutual exclusivity[26], resulting in repression of PAX7 transcriptional target genes in FSHD muscle, as we report.

Here, we show that when PAX7 and DUX4 are present in the same cell, the effect is the mutual inhibition of the respective transcriptional target genes of both proteins. Moreover,

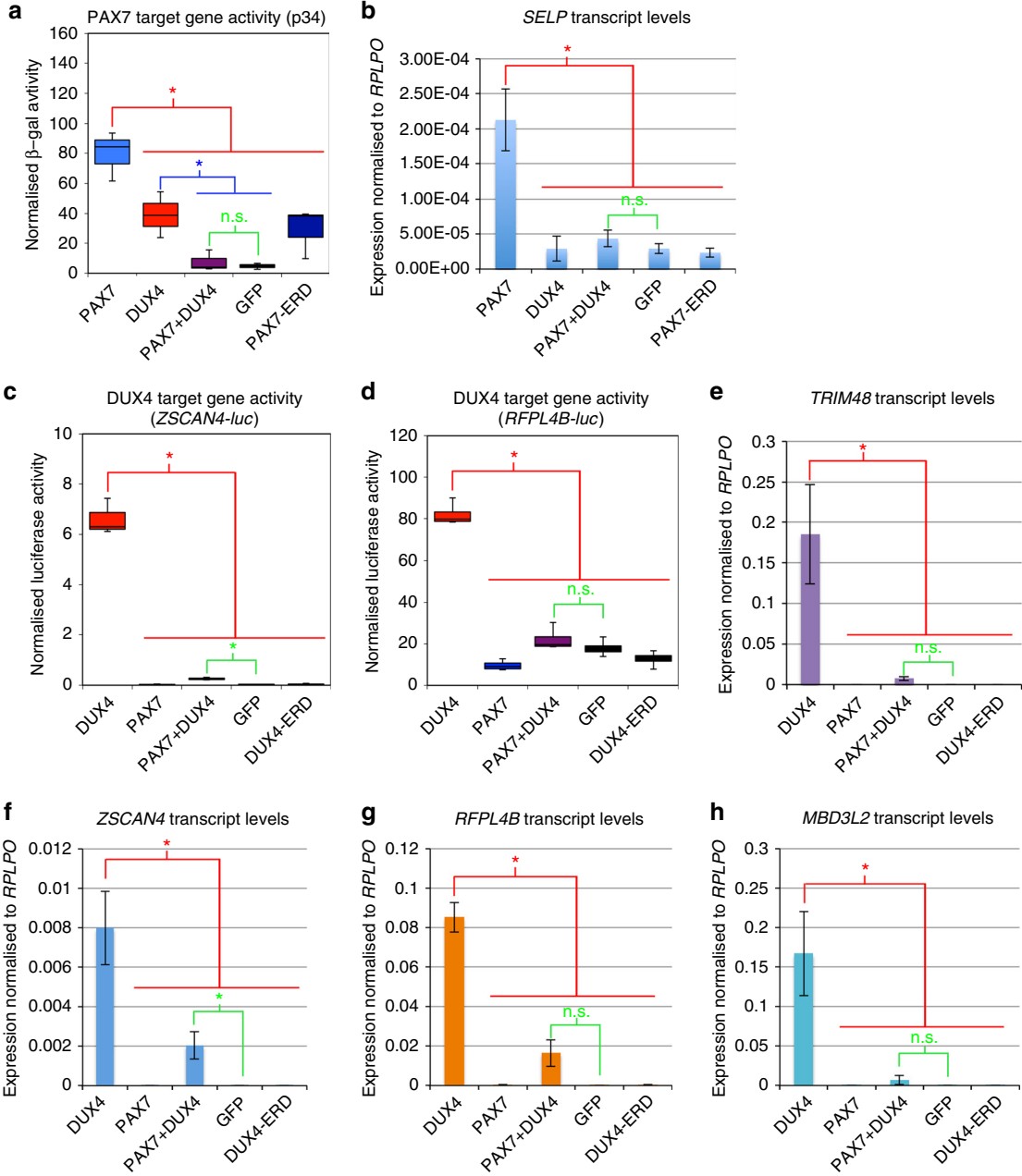

**Fig. 6** Co-expression of PAX7 and DUX4 causes suppression of the transcriptional activity of both proteins. HEK-293 or NIH-3T3 cells were transfected with plasmids encoding *DUX4*, *Pax7*, *DUX4* and *Pax7*, or dominant negative-version DUX4-ERD, or *Pax7*-ERD. **a** The ability of PAX7 to activate its transcriptional target genes was measured via the p34 plasmid driving *lacZ*, co-transfected with the constructs listed above, together with a control RSV luciferase plasmid as a transfection normaliser, into NIH-3T3 cells. Reporter gene intensities were measured using a Glomax-Multi + plate reader and normalised to RSV luciferase. PAX7 activated the p34 PAX7 reporter construct, but both PAX7 and DUX4 together suppressed activity of this PAX7 reporter compared to PAX7 alone. **b** RT-qPCR for PAX7 transcriptional target gene *SELP* confirms that co-expression of *Pax7* and *DUX4* suppresses the activation of this endogenous PAX7 transcriptional target compared to PAX7 alone, in HEK-293. The ability of DUX4 to activate its transcriptional target genes was measured via two separate DUX4 reporter constructs: **c** *ZSCAN4-luc* and **d** *RFPL4B-luc*, both controlling a luciferase reporter gene, co-transfected with DUX4 and/or *Pax7* constructs, or DUX4-ERD or GFP, together with a *RSV lacZ* as a transfection normaliser, into HEK-293. Reporter gene intensities were measured using a Glomax-Multi + plate reader and normalised to β-galactosidase activity. While DUX4 activated both DUX4 reporters, both PAX7 and DUX4 together, suppressed activity of these DUX4 reporters compared to DUX4 alone. RT-qPCR for DUX4 endogenous transcriptional target genes. **e** *TRIM48*, **f** *ZSCAN4*, **g** *RFPL4B* and **h** *MBD3L2* confirms that the presence of both PAX7 and DUX4 together suppresses activation of all these endogenous DUX4 transcriptional target genes compared to the levels achieved by DUX4 alone in HEK-293 cells. Boxes represents the interquartile range (IQR), with the median indicated by a line. Whiskers denote min (1.5*IQR, max (observed value)). For bar graphs, error bars denote standard error of the mean, $n = 3$ or 4 for each cell line, ANOVA revealed significant intensity differences, post hoc unpaired two tailed *t*-tests were employed to assess significant pairwise differences: *denotes $p < 0.05$, and n.s. denotes non-significance of pairwise *t*-tests

suppression of PAX7 transcriptional target genes when DUX4 and PAX7 are co-expressed is greater than would be expected under a competitive inhibition model. There are several possible mechanisms by which DUX4 and PAX7 proteins may interact to cause a suppression of PAX7 target genes in FSHD in addition to competitive inhibition. One is a direct protein–protein interaction, such as dimerization, which could disrupt target gene activation. Since PAX3 and PAX7 bind DNA as cooperative dimers[14, 51] via motifs in the homeobox domain[52], the homeobox domains of DUX4 may interfere with such dimerization to disrupt DNA-binding.

DUX4-mediated suppression of PAX7 transcriptional target genes, coupled with the report that PAX7 target genes can induce long-term epigenetic changes that persist in differentiated muscle[27], may explain our finding that PAX7 target gene repression is a feature of FSHD skeletal muscle. Such repression may directly contribute to FSHD molecular pathology. Indeed, we show here that PAX7 target gene repression is associated with over-expression of EZH2 target genes, an important pathway in epigenetic modification in skeletal muscle. EZH2 is a well-known epigenetic transcriptional repressor and component of PRC2 complex, which plays critical roles in development and cancer[53]. It has been shown that EZH2 is enriched at the D4Z4 region in healthy control primary myocytes, but not in FSHD patient-derived cells, contributing to epigenetic de-repression of DUX4 in FSHD[41]. Our data suggests that DUX4 inhibition of PAX7, resulting in perturbation of EZH2 targets, may also contribute to epigenetic de-repression of DUX4 in FSHD[54].

We further show that PAX7 target gene repression is associated with over-expression of HIF1α-mediated hypoxic response genes. Oxidative stress sensitivity is a well-known phenotype of FSHD myoblasts[54, 55], however, the precise mechanism is unclear. Studies focusing on DUX4 target gene expression have implicated perturbation of the glutathione redox pathway[11], while studies investigating FSHD patient-derived tissue in an unbiased manner have implicated HIF1α transcriptional dysregulation[29, 50]. Our results suggest that this latter mechanism could be attributed to DUX4-mediated inhibition of PAX7 target genes.

Mechanistically, HIF1α plays a complex role in cellular response to hypoxia and can be either pro- and anti-apoptotic, depending on oxygen levels, cell type and experimental set-up[56]. In normoxia, HIF1α undergoes proteasomal degradation, following oxygen-dependent ubiquitination by the Von Hippel Lindau tumour suppressor[57]. Hypoxia thus results in nuclear accumulation of HIF1α, where it dimerizes with HIF1β to form a transcription factor. HIF1α upregulation, following transient, mild levels of hypoxia is typically associated with cell survival and proliferation[56]. During sustained hypoxia, however, HIF1α induces apoptosis via a p53-dependent mechanism[58]. In FSHD myoblasts, DUX4 inhibition of PAX7, resulting in HIF1α upregulation, may mimic a sustained hypoxic response, resulting in increased p53-mediated apoptosis.

To conclude, our findings indicate that DUX4 likely operates in two ways to cause FSHD. Firstly, by driving the expression of a cohort of transcriptional target genes, many of which are pro-apoptotic, DUX4 can cause muscle pathology and cell death[59]. Secondly, by perturbing the ability of PAX7 to activate/maintain its transcriptional target genes, or by interfering with epigenetic changes elicited by PAX7, DUX4 represses PAX7 transcriptional target genes, likely causing pathology by mechanisms such as HIF1α over-activation and aberrant epigenetic changes. Thus, we propose that the model of FSHD molecular pathogenesis should be updated to incorporate PAX7 target gene repression as an important molecular mechanism and to refine current therapeutic strategies.

## Methods

**Published gene expression data.** Publicly available gene expression data used in this study fell into two distinct classes: data sets describing over-expression of Pax7 and DUX4, and data sets describing FSHD and control muscle biopsies. These ten independent studies encompassed 222 samples: four studies[10, 14, 33–35], totalling 35 samples described Pax7 and DUX4 over-expression. Six studies[3, 10, 28, 30–32], totalling 187 samples described FSHD and matched control muscle biopsies. Study references, public database accession numbers, experimental design descriptions, gene expression assay used and sample counts for each study are provided in Supplementary Data 6. For the RNA-seq FSHD muscle biopsy study published by Yao et al.[10], we removed control sample C6 from our analysis as it was derived from tibialis anterior, whereas all other samples in that study were isolated from quadriceps.

Publicly available microarray studies were obtained log normalised, from the GEO database. Quantile normalisation was subsequently performed across all samples within each study. To enable comparison across data sets and evaluation of the Pax7 and DUX4 scores, probes in each microarray data set and sequences in the RNA-seq data were matched to unique gene identifiers (EntrezGene for human and Ensemble for mouse). Probes or sequences mapping to the same gene identifier were averaged.

**Microarray of Pax7 constructs.** Procedures were carried out under the Animals (Scientific Procedures) Act 1986, as approved by King's College London Ethical Review Process committee. Satellite cells were obtained from myofibres isolated from the extensor digitorum longus (EDL) muscles of three adult (6–8-week-old) male C57BL/10 mice. Mice were killed by cervical dislocation and the EDL muscles removed intact and incubated in 0.2% Collagenase Type 1 in Dulbecco's modified Eagle medium (DMEM)-GlutaMAX (Invitrogen) at 37 °C for 90 min. Myofibres were then dissociated in DMEM-GlutaMAX under a stereo dissecting microscope using a series of heat-polished glass Pasteur pipettes and rinsed in changes of DMEM-GlutaMAX[60]. Intact myofibres were plated on serum-reduced Matrigel (BD) coated plates in proliferation medium (DMEM-GlutaMAX with 30% fetal bovine serum (PAA), 10% horse serum (Gibco), 1% chick embryo extract (ICN Flow), and 1% penicillin/streptomycin (Sigma) at 37 °C and 5% $CO_2$[60]. Expanded satellite cell-derived myoblasts were transduced with retroviral constructs encoding *Pax7* and dominant negative *Pax7-ERD*[15] or control retroviruses using 4 µg/ml Polybrene for 48 h. PAX7 is highly conserved, with 97% amino-acid sequence similarity between mouse and human. RNA was isolated using RNeasy-mini columns (Qiagen), processed using WT Gene Expression (Ambion) and GeneChip v4 Whole Transcript Sense Target Labelling Kit as per the manufacturers' instructions, and hybridised to Mouse Gene Chip 1.0 ST arrays using the GCS3000 microarray system (Affymetrix). Array data were normalised using robust multi-chip average (RMA), MvA transformation, and processed and statistically analysed using Partek Genomics Suite software (Partek Inc.). Principal component analysis on the 1000 most variable probes confirmed validity of biological replicates.

**Differential expression analysis.** Differential gene expression analysis was performed within two DUX4 over-expression studies[17, 35] and our Pax7 construct over-expression study to derive target gene sets. For microarray studies an empirical Bayes approach was employed utilising the limma package in R[61], while for RNA-seq data the DESeq2 package[62] was used to assess fold change. For DUX4 studies, to replicate the approach used in previous analyses[10], genes, which showed log FC > 2 and FDR < 0.05 were considered significantly upregulated by DUX4. DUX4 target genes from the Choi et al.[34] RNA-seq data and Geng et al.[35] microarray data are displayed in Supplementary Data 1 and 2. For our Pax7 microarray study a slightly different approach was employed to take advantage of the data from the dominant negative Pax7-ERD construct. Genes that showed both a log FC > 1 and $p < 0.05$ under Pax7 transduction and a log FC < 1 and $p < 0.05$ under Pax7-ERD transduction, were considered induced by Pax7. Genes that showed both a log FC < 1 and $p < 0.05$ under Pax7 transduction and a log FC > 1 and $p < 0.05$ under Pax7-ERD transduction, were considered repressed by Pax7: listed in Supplementary Data 3.

**Derivation and validation of Pax7 and DUX4 target genes.** Three DUX4 target gene scores were defined corresponding to DUX4 activated targets derived from the Yao et al.[10], Choi et al.[34] and Geng et al.[35] DUX4 over-expression. For each study, a single sample DUX4 target gene score was defined as the average expression of the DUX4 activated targets in a given sample.

Our Pax7 target gene score was defined from our Pax7 construct microarray derived induced and repressed targets (described above). A single sample Pax7 score was then defined as the *t*-statistic comparing the expression of induced and repressed Pax7 targets within the sample, analogous to an approach we previously employed[63, 64]. We validated our Pax7 score on a published data set[14], demonstrating that the score was significantly elevated in Pax7-expressing samples ($p < 5 \times 10^{-4}$, Supplementary Fig. 2).

Scores were z-normalised within each study, to reduce study-dependent effects, and evaluation of score differences between FSHD and control samples in a single study was performed via a Wilcoxon U-test.

**Meta-analysis of target gene scores**. Meta-analysis to assess the discriminatory powers of each of the target gene scores across the five microarray FSHD and control muscle biopsy data sets, was performed using a random effects model to combine statistics across studies. The p-values denoting the significance of the scores on meta-analysis were derived from a Fisher's combined test.

ROC curve analysis was performed using the pROC package in R[65].

**GSEA of the target gene scores**. GSEA was performed using a Fisher's exact test to compare Pax7 target gene sets against the gene sets defined by the Molecular Signatures Database[37, 38]. Computations were performed via software downloaded from the Molecular Signatures database (http://www.broadinstitute.org/msigdb).

**Human myoblast cell culture**. Five immortalised human myoblast cell lines from a mosaic FSHD1 patient 54-6, 54-A10 (control, 13 D4Z4 repeats) and 54-12, 54-A5, 54-2 (FSHD, 3 D4Z4 repeats)[45], were kind gifts from Dr. Vincent Mouly (Center for Research In Myology, UMRS 974 UPMC-INSERM, FRE 3617 CNRS, Paris, France) and Professor Silvère der Maarel (Leiden University Medic'al Center, Leiden, The Netherlands). Four immortalised human myoblast lines from two FSHD1 patients and two sibling matched controls: 12Ubic, 12Abic, 16Ubic and 16Abic, were obtained from Professor Charles Emerson (UMMS Wellstone centre for FSHD, Worcester, MA, USA)[44]. Immortalised human myoblasts were cultured in Skeletal Muscle Cell Growth Medium (Promocel) supplemented with 20% fetal bovine serum, 50 µg/ml Fetuin (bovine), 10 ng/ml epidermal growth factor (recombinant human), 1 ng/ml basic fibroblast growth factor (recombinant human), 10 µg/ml insulin (recombinant human), 0.4 µg/ml dexamethasone and 50 µg/ml gentamycin, at 37 °C under 5% $CO_2$. RNA was harvested from proliferating myoblasts as they reached confluency for each line in triplicate using RNeasy kit (Qiagen) according to the manufacturer's recommendation, including an additional DNA removal step using RNase-Free DNase Set (Qiagen).

**RNA seq and alignment**. RNA-seq libraries were prepared using the Agilent sureselect stranded RNAseq protocol, which allows polyA selection but was modified to work with ribodepletion. Prepared libraries were sequenced on an Illumina HiSeq2500. The raw reads were trimmed using trim-galore, utilising cutadapt[66] (v0.4.0) to remove the Illumina Sequencing Adapter (AGATCGGAAG AGC) at the 3′-end. Additionally, 12 bases were also trimmed from the 5′-end of the reads since they showed a biased distribution of bases. The reads were mapped to the human transcriptome using the human genome sequence GRCh38 and v82 gene annotations downloaded from Ensembl. The mapping was performed using tophat[67] (v2.1.0) and bowtie[68] (v1.1.0), enabling the fr-firststrand option of tophat to restrict mapping to the sense strand of the transcript. Reads were assigned to genes using the featureCounts program[69] (v1.5.0), counting fragments and ignoring multi-mapping reads. Gene assignment was also restricted so to the sense strand of the transcript. The resulting matrix of read counts was then loaded into R and normalised using DESeq2[62] before computation of Pax7 and DUX4 target gene scores.

**Reporter gene assays**. HEK-293 or NIH-3T3 cells (12,500 cells/well) were transfected with pMSCV-IRES-eGFP plasmids encoding DUX4 and/or Pax7, or dominant negative DUX4-ERD, or dominant negative Pax7-ERD[15, 29], together with reporter plasmids encoding either DUX4-responsive promoters driving luciferase expression (pZSCAN4-luc[35] or RFPL4B-luc) or a Pax3/7-responsive element driving β-galactosidase (p34 plasmid[15]) using Lipofectamin LTX (Thermofisher). A pRSV vector encoding either β-galactosidase or luciferase as appropriate was co-transfected as an internal control for normalisation of transfection efficiency. Total amount of DNA transfected per reaction was 1.5 mg, in equal ratios for reporters and plasmids encoding genes of interest, and 250 ng/reaction of the internal normaliser vector. Cells were harvested 24 h after transfection, and assayed using the Dual-light Reporter system (Thermofisher) according to the manufacturer's instructions. 3-4 independent transfections were performed in technical triplicate. Reporter gene activity was measured using Glomax-Multi + plate reader (Promega) and normalised for transfection efficiency. $N = 3/4$ for each cell line, analysis of variance (ANOVA) revealed significant intensity differences, post hoc unpaired two-tailed t-tests were employed to assess significant pairwise differences.

**Reverse transcription quantitative PCR**. HEK-293 (150 000 cells/well) were transfected with plasmids encoding DUX4 and/or Pax7, or dominant negative DUX4-ERD, or dominant negative Pax7-ERD[15, 29]. RNA was isolated after 24 h and reverse-transcribed using the Reverse Transcription Kit with genomic DNA wipeout (Qiagen); RT-qPCR was performed on a Viia7 qPCR system (Life Technologies) with MESA Blue qPCR MasterMix Plus and ROX reference dye (Eurogentec). Primers used were as follows:

RPLPO (FWD: 5′-TCTACAACCCTGAAGTGCTTGAT-3′, REV: 5′-CAATC TGCAGACAGACACTGG-3′)

ZSCAN4 (FWD: 5′-TGGAAATCAAGTGGCAAAAA-3′, REV: 5′-CTGCATG TGGACGTGGAC-3′)

TRIM48 (FWD: 5′-TGAATGTGGAAACCACCAGA-3′, REV: 5′-GTTGAGC CTGTCCCTCAGTC-3′)

RFPL4B (FWD: 5′-GAGACGTAGGCTTCGGATCTT-3′, REV: 5′-GGCTGA ATTCAAGTGGGTCT-3′)

MBD3L2 (FWD: 5′-GCGTTCACCTCTTTTCCAAG-3′, REV: 5′-GCCATGT GGATTTCTCGTTT-3′)

SELP (FWD: 5′-CGCCTGCCTCCAGACCATCTTC-3′, REV: 5′-CTATTCA CATTCCAGAAACTCACCACAGC-3′)

**Data availability**. Microarray and RNA-seq data are available from the GEO data base (https://www.ncbi.nlm.nih.gov/geo/), accession numbers GSE77478 (microarray) and GSE102812 (RNA-seq).

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

## Acknowledgements

We are grateful to colleagues who generously shared cells lines, including Dr. Vincent Mouly (Center for Research In Myology, UMRS 974 UPMC-INSERM, FRE 3617 CNRS, Paris, France), Professor Silvère der Maarel (Leiden University Medical Center, Leiden, The Netherlands) and Professor Charles Emerson (UMMS Wellstone centre for FSHD, Worcester, MA, USA). We heartily thank Dr. Stephen Tapscott (Fred Hutchinson Cancer Research Center, 1100 Fairview Ave. N. Seattle, WA 98109-1024) for providing the DUX4 reporter constructs. We are grateful to Efthymios Fidanis and Dr. Alka Saxena of the Genomics Research Platform, Biomedical Research Centre at Guy's and St Thomas' Trust and Kings College London for Library preparation and Sequencing: supported by the National Institute for Health Research (NIHR) Biomedical Research Centre based at Guy's and St Thomas' NHS Foundation Trust and King's College London. C.R.S.B. was supported by CoMPLEX Ph.D studentship and the British Heart Foundation (SP/08/004) and then the FSH Society (FSHS-82016-03). S.S. is supported by the Royal Society, the EPSRC and the National Natural Science Foundation of China. R.B.W. received funding from Muscular Dystrophy UK (RA3/762) and The Wellcome Trust (085137/Z/08/Z). M.P. is funded by Muscular Dystrophy UK (RA3/3052/1). H.H. was supported by the British Heart Foundation (PG/13/1930059). We are particularly thankful to the FSH Society Shack Family and Friends research grant (FSHS-82013-06), The King's Health Partners Research and Development Challenge Fund (R151006) and Association Francaise contre les Myopathies (17865). The Zammit laboratory is additionally supported by the Medical Research Council (MR/P023215/1), together with BIODESIGN (262948-2) from EU FP7.

## Author contributions

The overall study was conceived by C.R.S.B. and P.S.Z. PAX7 microarray data was generated during a project conceived and directed by P.S.Z. and F.R. Statistical analysis

was performed by C.R.S.B. Experiments were performed by C.R.S.B., M.P. and R.B.W. RNA-Seq data was pre-processed and mapped by C.R.S.B. and H.H. and analysed by C.R. S.B. The manuscript was mainly written by C.R.S.B. with input from P.S.Z. and contributions from M.P. and S.S. and proof-read by all authors.

## Additional information

**Competing interests:** The authors declare no competing financial interests.

**Change history:** A correction to this article has been published and is linked from the HTML version of this paper.

