## [Peer Review File · Nature Communications]

Reviewers' comments:

Reviewer #1 (Remarks to the Author):

The main question raised in this manuscript is to determine if in FSHD, DUX4 directly activates a set of downstream genes or if DUX4 inhibits the binding of PAX3/7 to genomic DNA, leading to a repression of the genes downstream of PAX3/7. The question has not been addressed before and is interesting because homologies between the DNA binding domains of DUX4 and PAX3/7 have been previously reported.

The authors have first reanalyzed several microarray data series to identify target genes of DUX4, PAX3 and PAX7. This step is essential to determine if the expression of PAX7 and/or DUX4 target genes is altered in FSHD compared to control. The authors have chosen several articles in the literature for this first analysis and I have several concerns about their choices. For instance, in the Geng article, primary control cells were transduced with a lentiviral vector expressing DUX4-fl and analyzed 24 h later. But in the Barneji article, primary mouse satellite cells from C57BL10 mice were transduced with a DUX4 expressing retrovirus and analyzed 20h later. In the Bosnakovski article, C2C12 cells stably carrying an inducible DUX4 were used and the results were analyzed up to 12h after DUX4 induction. The methodology is different in all these articles. More importantly, the authors compare data generated from mouse and human cells. This last point is very important since it has been demonstrated in 2013 (Young et al, Plos Genet) that DUX4 binding to DNA creates promoters that are active in FSHD muscles. Because the non-coding regions are particularly subject to DNA modification between the human and murine genome, an identical set of activated genes is not expected in human and mouse cells. This was actually partially demonstrated in 2013 by Sharma et al who have shown that DUX4 differentially regulates transcriptomes of human rhabdomyosarcoma and mouse C2C12 cells (Plos One).

The approach generating a single sample score is thus problematic for DUX4. Because of the choices made to identify the DUX4 target genes, I don't know how to interpret the results. It would be of interest to know the overlap between the set of genes that was identified to correlate with DUX4 in the human study with the corresponding sets in the murine studies. It would also be of interest to know whether a sample score based only on the human study would give similar results as the sample score based on the combination of the human and murine studies.

Furthermore, to investigate the activity of the genes downstream of DUX4, PAX3 or PAX7 in FSHD biopsies, the authors have used 5 published microarray studies but some other studies (for example, Celegato et al, 2006, and Winokur et al 2003, at least) have not been selected. An explanation should be provided of why certain data series were included and others were not.

Are immortalized cells the best model to globally analyze repressed and activated genes? It is known that immortalization changes the methylation status of genomic DNA and can generate aneuploid cells. In both cases, gene expression could have been affected by the immortalization and the authors compare only 1 FSHD and 1 control clone. Did the authors check these points? Have they performed the same experiments on other immortalized clones (5 FSHD and 5 control clones have been analyzed in the previous article describing these clones and differences were reported).

As the work features many qPCR experiments, it would be ideal if the Authors could follow the MIQE standards to improve the power of their findings. Methodology is not detailed. What is the normalizer? How did the authors choose it? What kind of statistical test was done to choose the normalizer? If the normalizer is not chosen properly, all the results are subject to change.

In figure 2, how was the statistical analysis performed? Did they compare FSHD myoblasts and control myoblasts on one hand and FSHD myotubes and control myotubes on the other hand or were the two cell states in fact combined (as is suggested by bars indicating statistical significance)? What happens if a MANOVA statistical analysis is done?

In figure 3, did the authors compare Pax7 transfected cells versus all the other conditions at the same time? The experimental method is poorly described. What is p34 exactly? What is the DUX4 expression plasmid used? What quantity of plasmids was used for the transfection?

A T-test is not appropriate in this figure.

Why did the authors chose to transfect a plasmid expressing murine Pax7 instead of Human PAX7?

The results shown in Figure 3A are intriguing: Pax7 over-expression induces p34 reporter activity. Similar results are obtained when DUX4 is over-expressed. However, when both DUX4 and Pax7 are expressed, no change in reporter activity is observed. This needs to be discussed. What could be the possibilities having in mind the homologies between the DNA binding domains of DUX4 and PAX3/7?

In figure 3B, the co-transfection of DUX4 and Pax7 does not activate the DUX4 reporter construct. But in Fig 3C, it does. How do the authors explain this difference? The conclusion of the authors concerning these experiments is that co-expression of Pax7 and DUX4 acts to suppress the transcriptional targets of both DUX4 and Pax7, which is obviously not the case (fig 3C).

The GSEA analysis is of interest but the authors could examine more closely the genes involved, for example using Leading Edge Analysis (Leading Edge genes are those that drive the up- or down-regulation of the gene sets - technically they are identified by GSEA to lie before the maximum of the running enrichment score). For example, it is curious that only one Hypoxia-related gene set out of 48 was significantly enriched - which genes drive this enrichment? Are they a subset of genes that are special to that hypoxia-related gene set that are not found in other hypoxia-related gene sets? How do these hypoxia-related leading edge genes compare to those hypoxia-related genes that the authors identified to be dysregulated in their previous work? As a second example, it is very interesting that Hypoxia and EZH2 targets are both enriched in both PAX3 and PAX7 - do the leading edge genes of these two gene sets overlap with each other? Do they overlap between the PAX3 and PAX7 studies? The two examples relate to a more basic question that the authors could address: is there a small subset of genes that are strongly regulated by PAX3 and PAX7 (a subset that overlaps with Hypoxia and with EZH2 targets)? It would also be of interest to see the GSEA enrichment curves for these two gene sets for PAX3 and PAX7.

In their conclusions, the authors declare that DUX4 expression inhibits both proliferation and myogenic differentiation, a phenotype similar to that observed when PAX3/7 expression is repressed. However, in the Krom et al article describing the immortalized clones, the fusion index was improved in the FSHD clones. How do the authors explain that?

Finally, considering that (1) most of the articles using primary FSHD cells have shown that DUX4 expression is barely expressed in FSHD myoblasts whereas it is highly expressed in FSHD myotubes and (2) satellite cells account for 2-7% in adult muscles, and (3) PAX7 is only expressed in satellite cells, how can it be that suppression of PAX3/7 transcriptional targets is the hallmark of FSHD? The discussion about low and high levels of DUX4 in muscle fibers is interesting and should be more developed to incorporate these points. Where could be the most important role of DUX4 in the physiopathology of FSHD: in the muscle fiber (leading to its apoptosis) or in the satellite cells (inhibiting myoblast differentiation)? Is there any link between the number of satellite cells per fiber and FSHD affected muscles?

Reviewer #2 (Remarks to the Author):

Zammit and colleagues address an important question in the field of FSHD, namely the lack of consistency in transcriptional profiling data in cells and tissue samples from FSHD patients. Although the DUX4 gene is known to cause FSHD, the mechanism by which it acts to cause muscle pathology is unknown. At high levels of expression, DUX4 induces a group of genes, collectively

described as the DUX4 signature. Zammit and colleagues look for this signature in a large set of published data and do not find it. This is a very provocative and important result.

Rather, they find an inverse PAX7 signature, meaning that genes that are regulated by Pax7 are found to be inversely regulated in FSHD samples. The authors speculate that this may be an effect of low level DUX4 expression, distinct from the DUX4 signature seen with high level expression, in which DUX4 antagonizes PAX7. The connection to PAX7 is quite interesting as the two proteins have similar homeodomains and it has been shown that PAX7 and DUX4 can compete with one another.

This study is an important contribution to the field and will be received with great interest. There are however a few issues that should be addressed:

1. The authors evaluate expression in immortalized isogenic myoblast cell lines. Although the original study describing these lines reported derivation of multiple FSHD and control clones from a mosaic individual, and indeed drew conclusions from evaluating multiple clones, the authors only use a single clone. As the previous paper showed significant inter-clonal variation, data from comparing two groups of n=1 clone is not reliable. The authors should repeat this part of the study using the full panel of isogenic clones.
2. In Figure 2, the DUX4, PAX3 and PAX7 scores are shown for myotubes and myoblasts for FSHD and control lines. Shouldn't myotubes have a lower PAX3 and PAX7 score than myoblasts? They ought to have no or low levels of Pax7. Yet the levels are similar. Please explain.
3. A further concern about the PAX3/7 signatures is that they may be detecting "quality" of myoblasts. Perhaps myoblast cultures from diseased muscle simply show a reduced score on this metric, perhaps because they tend to be more prone to differentiate or are more prone to fibroblastic contamination. To address this, please evaluate the PAX3/7 signature on independent datasets, specifically diseased vs. control myoblasts from published Duchenne MD profiling studies.
4. Cotransfection experiments can be subject to artifact related to efficiency of DNA uptake. Can Figure 3 be supported with data on endogenous genes, in addition to the transfected reporters? I.e., in addition to looking at the co-transfected ZSCAN4 reporter, what is the level of the endogenous ZSCAN4 gene? Of some endogenous Pax7 target genes.
5. The authors only evaluate DUX4 and PAX3/7 signatures. It would be valuable to test some negative control signatures. How significant is the inverse Pax7 signal vis a vis other potential signals? I.e. if you selected lineage-specific transcription factors at random, would they show correlations in the FSHD data.
6. Along the same lines, it has been shown that DUX4 has effects on MYOD. Can the authors evaluate the MYOD signature to determine whether FSHD vs control samples show alterations in this?

Reviewer #3 (Remarks to the Author):

A. Summary: The DUX4 gene, linked to FSHD, is a transcription factor. A number of mechanisms by which DUX4 could contribute to FSH muscular dystrophy have been proposed and are under investigation in the field. Most of these mechanisms involve DUX4 activation of target genes that have been implicated in a number of potentially deleterious pathways. An alternative mechanism, proposed several years ago by Bosnakovski, et al (reference 5 in the current manuscript) suggests that DUX4 acts by interfering with PAX3/PAX7-derived gene expression, and that PAX3/7 target

gene repression may play a role in FSHD. The current manuscript by Banerji, et al explores this hypothesis. To do this, they perform meta analysis of several published gene expression studies in which DUX4/DUX4-fusion proteins or PAX3/7 or PAX3/7-forkhead fusion proteins were over-expressed in various cells. They also performed an additional PAX7 / PAX7-fusion protein over-expression study themselves, to provide some new data to the analysis. They used these over-expression studies to generate DUX4-, PAX3- and PAX7-expression signatures, and then queried previously published gene expression datasets derived from FSHD cells/biopsies in which DUX4 was present (or presumably present, since it wasn't measured in many cases) at extremely low, endogenous levels. Based on these analyses, the authors conclude that DUX4 target genes are not changed in FSHD muscles, but that PAX3/7 target genes were significantly repressed, lending support to the hypothesis that inhibition of PAX3/7 target genes are a contributing factor to FSHD pathogenesis.

B. Originality and interest: From the standpoint of this study being a meta-analysis largely utilizing others' published data, a large amount of the actual data is not new. However, there have been a number of FSHD gene expression studies performed, as noted by the authors, and in studies that did not utilize DUX4 over-expression methods, there seemed to be little overlap in expression changes among the various published works. Nevertheless, a rigorous statistical comparison has not been published, to my knowledge. So an analysis like the one here is long overdue, and as such, a very welcome and important pursuit for the field.

C-F. The authors defined their DUX4, PAX3 and PAX7 expression signatures using datasets in which each respective protein was overexpressed. On this surface, this seems logical and sensible. However, there are a few issues with this approach that need consideration and could detract from the power of this analysis.

Issue 1: Differences in cell type and species used to generate gene expression signatures.

Specifically, the authors used three different datasets to generate their DUX4 data. Two were performed in two different kinds of mouse cells (mouse primary satellite cells, and mouse C2C12 myoblasts), while the third was performed in immortalized human myoblasts. These differences could have a large influence on gene expression changes, thereby calling into question the reproducibility of the data from one experiment to the next. Specifically, close DUX4 homologs are present only in primates. Mice do not contain DUX4, although they do express a paralogous gene (Dux) that is structurally similar to DUX4 but fairly divergent on the amino acid level. Certainly there are a number of studies supporting that mouse can be a useful model to study DUX4 activity, and this reviewer believes that the mouse has utility for studying various aspects of DUX4 and FSHD. However, considering the two genes have been under different selective pressures for millions of years, and there are no published data suggesting they have maintained the same gene targets, it is not clear to me that a DUX4 gene expression profile generated in mouse cells is comparable to one obtained in human cells. Thus, the reproducibility of the DUX4 over-expression datasets used to generate DUX4 gene signatures comes into question. The species differences seem to be less of a concern with the PAX3/7 datasets since these genes are well-conserved in mammals, and even further back in lower organisms, such as zebrafish.

Issue 2: A second issue related to the over-expression datasets for generating gene-specific expression signatures is the use of fusion proteins. Specifically, the PAX3 and PAX7 datasets relied upon PAX3-forkhead, and PAX7-forkhead proteins derived from translocations found in rhabdomyosarcomas. Thus, the transactivation domains are different from the natural PAX3 and PAX7 proteins. Is this the same as PAX7 and PAX3? Is it legitimate to compare PAX3 and PAX7-derived datasets to ones utilizing these translocations (which are different proteins)? Moreover, one of the DUX4 datasets, and one of the PAX7 datasets, also used fusions to an ERD repressor domain, or in the case of DUX4, to a VP16 transactivation domain. Considering these C-terminal regions may contain binding sites for transcriptional co-factors, and they are different in many cases, is it possible that these different transactivation regions could influence gene expression

profiles, thereby making the determination of a PAX3-, PAX7-, or DUX4-specific expression profile less reliable? I am uncertain about this.

Issue 3: Performing a meta-analysis like the one here obviously relies upon others' published work. In this case, the authors of this manuscript compared 5 gene expression studies in which biological materials were analyzed using different expression arrays, from at least three different muscle groups, in different populations of FSHD patients. It was unclear, however, how many of these materials showed DUX4 expression, and if so, at what levels? Do the authors of this manuscript have this information? If so, it should be included in Suppl Table 1. Moreover, would the authors expect that the presence/absence of DUX4 would impact the presence/absence of a DUX4-target gene expression profile in a dataset? Likewise, would the presence/absence of DUX4 impact its ability to interfere with PAX3/7 genes?

Other comments related to the data and data presentation: as this type of analysis has not been performed before, I think it would be extremely useful to include a summary of each study used in this manuscript. Specifically, how many genes were on each microarray? How many total genes were significantly changed (up or down)? How many were in common between each individual study used here? How many differences? It would also be useful to have the same analysis done with a control transcription factor (preferably a homeodomain containing protein) that is unlinked to the FSHD pathogenic mechanism. Are similar gene expression data available for other HOX genes?

On lines 51-53 of the manuscript, the authors stated that "A single sample score for each transcription factor was constructed via comparison of activated and repressed target gene expression levels in a given sample, with genes in multiple transcription factor scores removed to ensure independence". Could the authors please clarify this methodology and rationale for doing so in greater detail, perhaps also by providing an example of how removing such genes impacts the score? How many genes were excluded? Is it worthwhile to include a list of excluded genes in the supplement?

The data in Figure 2, using the mosaic patient cells, are interesting, potentially important and seem to support the authors' meta analysis. The major drawback to these data are that they are derived from N=1 patient.

Finally, I'm uncertain that the title accurately reflects the paper. There are two issues related to my way of thinking here. First, considering the concerns laid out above (related to differences in cell type, species, and wild-type versus fusion proteins) is it completely accurate to lump the expression changes into bins of PAX3-, PAX7- and DUX4-target genes as stated in the title and throughout the manuscript? Second, if we put this argument aside, and focus solely on the data as presented, the authors show that DUX4 transcriptional targets are not observed, while PAX3/7 target genes are reduced. Considering this only, one could argue then that the title does reflect the data, but not the conclusions. Specifically, in the conclusions, the authors suggest that DUX4 likely operates by inhibiting PAX3/7 genes, and although their data do not show it, they suggest DUX4 could still contribute to FSHD pathogenesis by activating its own targets.

G: Appropriate.

H. Appropriate.

Response to Reviewer 1

Reviewer 1 – response to our initial reply to reviewers:

Reviewer Comment: I am not convinced by the data even if I think they are interesting. In the first set of experiments, when they identify DUX4 and Pax7 target genes, only 3 publications relative to DUX4 were used, among them 2 were performed on murine samples. And as I said, DUX4 binding to DNA creates promoters that are active in FSHD muscles. Because the non-coding regions are particularly subject to DNA modification between Human and mouse genome, a similar set of activated genes is not expected in human and mouse cells. Moreover (and as mentioned by Reviewer 3), the clusters are different when DUX4-VP16 RV or DUX4-ERD RV are used (response, Fig. 1A).'

'Pax7 plays a major role in both Human and murine muscle physiology and a conserved set of activated genes is expected. However, for DUX4, it may be different: the D4Z4 region does not exist in rodent and DUX4 is barely expressed in human control muscles. A selective pressure has not occurred to maintain a conserved set of DUX4 downstream genes. "

RESPONSE: We appreciate that over-expression of DUX4 in mouse may lead to an omission of DUX4 targets unique to human myoblasts. Thus, we have removed all mouse DUX4 over-expression data sets and no longer present Fig 1A. Instead, we have compared 3 DUX4 target gene signatures derived from transcriptomic data profiling human myoblasts expressing DUX4 and matched controls. Firstly, we consider the 114 robustly upregulated DUX4 genes described by Yao et al., 2014, derived from RNA-seq of DUX4 lentivirus transduced human myoblasts. Secondly, we consider 212 upregulated DUX4 target genes we have derived from data provided in Choi et al. 2016 corresponding to RNA-seq of iDUX4 human myoblasts (using the same significance criteria as described by Yao et al. 2014). Finally, in order to compare differences between DUX4 targets identified by microarray and RNA-seq we also considered 165 upregulated DUX4 target genes we have derived from data provided in Geng et al. 2012, corresponding to microarray of DUX4 lentivirus transduced human myoblasts.

First Round comments

Reviewer Comment: The main question raised in this manuscript is to determine if in FSHD, DUX4 directly activates a set of downstream genes or if DUX4 inhibits the binding of PAX3/7 to genomic DNA, leading to a repression of the genes downstream of PAX3/7. The question has not been addressed before and is interesting because homologies between the DNA binding domains of DUX4 and PAX3/7 have been previously reported.

The authors have first reanalyzed several microarray data series to identify target genes of DUX4, PAX3 and PAX7. This step is essential to determine if the expression of PAX7 and/or DUX4 target genes is altered in FSHD compared to control. The authors have chosen several articles in the literature for this first analysis and I have several concerns about their choices. For instance, in the Geng article, primary control cells were transduced with a lentiviral vector expressing DUX4-fl and analyzed 24 h later. But in the Barneji article, primary mouse satellite cells from C57BL10 mice were transduced with a DUX4 expressing retrovirus and analyzed 20h later. In the Bosnakovski article, C2C12 cells stably carrying an inducible DUX4 were used and the results were analyzed up to 12h after DUX4 induction. The methodology is different in all these articles.

More importantly, the authors compare data generated from mouse and human cells. This last point is very important since it has been demonstrated in 2013 (Young et al, PLoS Genet) that DUX4 binding to DNA creates promoters that are active in FSHD muscles. Because the

non-coding regions are particularly subject to DNA modification between the human and murine genome, an identical set of activated genes is not expected in human and mouse cells. This was actually partially demonstrated in 2013 by Sharma et al who have shown that DUX4 differentially regulates transcriptomes of human rhabdomyosarcoma and mouse C2C12 cells (Plos One).

RESPONSE: We appreciate the reviewer's concern on the difference between mouse and human identified DUX4 transcriptional targets. As discussed above, we now only consider DUX4 target gene identified in 3 human myoblast over-expression studies and omit the previous 2 mouse DUX4 studies.

Reviewer Comment: The approach generating a single sample score is thus problematic for DUX4. Because of the choices made to identify the DUX4 target genes, I don't know how to interpret the results. It would be of interest to know the overlap between the set of genes that was identified to correlate with DUX4 in the human study with the corresponding sets in the murine studies. It would also be of interest to know whether a sample score based only on the human study would give similar results as the sample score based on the combination of the human and murine studies.

RESPONSE: We appreciate that the approach used for our single sample DUX4 score was difficult to compare with previous studies and that a clear comparison of the derived biomarkers was not made making interpretation difficult. To remedy this we have re-performed all analysis to ensure score compatibility and include a robust comparison of biomarkers. As mentioned earlier we now only consider DUX4 over-expression in human myoblasts, so comparison with murine studies is no longer a limitation of the analysis.

For clear comparison with previous studies, we begin with 2 previously published FSHD biomarkers: Yao et al. 2014: the mean expression of 114 robustly induced DUX4 targets in a given sample and Rahimov et al. 2012: the mean expression of 15 FSHD associated transcripts in a given sample. We find that neither biomarker validates across multiple datasets. We then derive two new DUX4 target gene signatures from two separate human myoblast DUX4 over-expression studies (one microarray Geng et al. 2012 and one RNA-seq Choi et al. 2016). To facilitate comparison with previous studies we identify these targets in an identical way to that described by Yao et al. 2014 (DUX4 induced targets display $FDR < 0.05$, $\log FC > 2$). We provide the gene lists as Supplementary Tables 1 & 2 and comment on the over-lap between studies.

We note that in using the new Choi et al. 2016 RNA-seq DUX4 target signature we can separate FSHD samples from controls on the FSHD muscle biopsy data set profiled by RNA-seq, but not on data sets profiled by microarray. By contrast, the Geng et al. 2012 microarray derived DUX4 target gene signature is capable of discriminating FSHD from control muscle biopsies profiled by RNA-seq data and microarray data (though the later requires meta-analysis to show significance and no single muscle biopsy data set shows significant elevation of DUX4 targets on FSHD samples). As stated in the updated manuscript:

'These results indicate that DUX4 target gene expression is a weak but significant biomarker of FSHD status but points to a lack of compatibility between RNA-seq and microarray studies for evaluating DUX4 transcriptional targets. Despite overlap between the 3 DUX4 signatures (28 genes shared between Geng et al. 2012 and Choi et al. 2016, 45 between Yao et al. 2014 and Geng et al. 2012 and 29 between Choi et al. 2016 and Yao et al. 2014), we see that RNA-seq derived DUX4 up-regulated targets are only capable of discriminating FSHD biopsies from controls if the biopsies were also profiled by RNA-seq. In contrast, microarray based DUX4 up-regulated targets show discriminatory power on both microarray and RNA-seq datasets. This is likely due to differences in coverage and dynamic range

across the two technologies and is a consideration when evaluating DUX4 transcriptional targets on FSHD samples.'

Our derivation of the Pax7 score is slightly different to the DUX4 targets as detailed in the manuscript:

'To derive a set of PAX7 targets we assayed primary murine satellite cell-derived myoblasts over-expressing Pax7, a dominant negative version Pax7-ERD and control retroviral constructs. We assayed gene expression via microarray to facilitate compatibility with the publically available FSHD muscle biopsy data sets. Microarray data was pre-processed and normalised as described in Methods, hierarchical clustering and principal component analysis, confirmed reproducibility of the transcriptional landscapes induced by the PAX7 constructs and close clustering of replicates.

Unlike DUX4 which is a potent transcriptional activator, PAX7 likely modulates transcription in more complex ways. Hence, to derive a biomarker of Pax7 expression in murine satellite cells, rather than focus only on the most strongly induced target genes, we performed differential expression analysis to derive a set of 311 up-regulated target genes (defined as induced by Pax7 over-expression and suppressed by Pax7-ERD) and a set of 290 down-regulated target genes (defined as suppressed by Pax7 over-expression and induced by Pax7-ERD) (Supplementary Table 3). A biomarker of Pax7 was then defined from consideration of the ratio of mean up-regulated to down-regulated target expression in a given sample (Methods). To validate this ratio as a biomarker of Pax7 expression, we evaluated it on an independent microarray data set describing Pax7 retroviral over-expression in primary murine satellite cells alongside control retrovirus in triplicate¹⁴. Our Pax7 biomarker demonstrated significantly higher values on samples over-expressing Pax7 in this independent data set, so confirming validity (Supplementary Fig. S2).'

'We next evaluated our Pax7 biomarker on the 6 FSHD muscle biopsy datasets. We found that the levels of Pax7 targets were significantly repressed in FSHD muscle biopsy samples profiled by RNA-seq (Fig. 3B). Importantly, Pax7 target repression was also found to be a significant biomarker of FSHD status in each of the 5 microarray biopsy studies independently, leading to a highly significant repression of Pax7 targets on meta-analysis ($p=3.5 \times 10^{-9}$, Fig 3C).'

'The Pax7 and DUX4 biomarkers were derived differently, so it is important to consider how the results would change if this were not the case. Firstly, if repressed DUX4 targets are combined with the induced targets to construct a biomarker of DUX4 analogous to our derivation of the Pax7 biomarker, DUX4 target genes as a discriminator of FSHD status are lost. This indicates that only robustly upregulated DUX4 targets have discriminatory power in FSHD and that suppressed targets introduce noise which masks this signal. Conversely, if we separately consider the up-regulated or down-regulated targets of Pax7, rather than together as a biomarker of Pax7 (analogous to the DUX4 biomarker), we see that up-regulated targets are significantly repressed in FSHD microarray samples on meta-analysis, whilst down-regulated targets are significantly over-expressed, confirming that both activated and suppressed Pax7 targets are perturbed in FSHD (Supplementary Fig. S4). However, the up or down regulated targets alone are unable to discriminate between FSHD and control muscle biopsies on the RNA-seq dataset.

This cross comparison demonstrates that the biomarker construction which maximises DUX4 target gene discriminatory power in FSHD focuses only on upregulated targets, whilst that which maximises Pax7 target gene discriminatory power utilises both up and down regulated target genes.'

A formal comparison of DUX4 and Pax7 biomarkers was performed utilising ROC curve analysis and De-Long's test to compare AUC for the discriminatory capacity of the different biomarkers. Due to data differences we do this separately on microarray and control samples. We find that our Pax7 biomarker is superior to DUX4 target gene expression as a signature of FSHD muscle on microarray muscle biopsy datasets, and as good a signature across RNA-seq data sets. Available RNA-seq data sets samples are more limited in number, and so the analysis is somewhat underpowered.

Reviewer Comment: Furthermore, to investigate the activity of the genes downstream of DUX4, PAX3 or PAX7 in FSHD biopsies, the authors have used 5 published microarray studies but some other studies (for example, Celegato et al, 2006, and Winokur et al 2003, at least) have not been selected. An explanation should be provided of why certain data series were included and others were not.

RESPONSE: The Celegato et al, 2006 study was omitted as it uses competitive array hybridisation of FSHD and controls rather than profiling controls and FSHD samples separately, unfortunately rendering this data incompatible with our meta-analysis. The same data set presented in Winokur et al 2003 was used in Bakay et al. 2006 as it was published twice, we have referenced Bakay et al. rather than Winokur et al. as the data link cited in Winokur et al. is no longer active, whereas that cited by Bakay et al. is maintained by GEO, hence the latter is more appropriate for locating the data to reproduce our analysis.

Reviewer Comment: Are immortalized cells the best model to globally analyze repressed and activated genes? It is known that immortalization changes the methylation status of genomic DNA and can generate aneuploid cells. In both cases, gene expression could have been affected by the immortalization and the authors compare only 1 FSHD and 1 control clone. Did the authors check these points? Have they performed the same experiments on other immortalized clones (5 FSHD and 5 control clones have been analyzed in the previous article describing these clones and differences were reported).

RESPONSE: We appreciate this point and have now validated the suppression of Pax7 target genes in 5 FSHD patient lines and 4 control lines as detailed in the updated version of the paper as follows:

'We performed RNA-sequencing on immortalised human myoblasts isolated from 3 independent FSHD patients alongside matched controls. One of the FSHD patients considered is mosaic for the FSHD genotype and multiple clones are available, of which we considered 5. These clones are isogenic with exception of the D4Z4 region, which is truncated to 3 repeats in clones 54-12, 54-2 and 54-A5 (an FSHD genotype), whereas clones 54-6 and 54-10 have 13 repeats (healthy number). The two further FSHD patient cell lines considered were the 12A and 16A immortalised myoblasts obtained from the Wellstone centre alongside sibling matched controls. RNA-sequencing was performed in triplicate on confluent myoblasts for each cell line, giving a total of 15 FSHD samples (3 patients corresponding to 5 lines in triplicate) and 12 control samples (3 patients corresponding to 4 lines in triplicate).

Our Pax7 biomarker alongside the Yao et al. 2014, Choi et al. 2016 and Geng et al. 2012 DUX4 target gene signatures were computed for each sample and scores for each patient and matched control were z-normalised and pooled for analysis. We found that our Pax7 biomarker, the Yao et al. 2014 and Geng et al. 2012 DUX4 target gene signatures were all significant discriminators of FSHD status (Wilcoxon $p < 0.05$, Fig. 5A), in line with the results from the muscle biopsy RNA-seq data, however the Choi et al. 2016 DUX4 target gene signature was not. ROC curve analysis demonstrated that there was no significant difference between the discriminatory power of our Pax7 biomarker and the DUX4 biomarkers on FSHD patient cell lines (De-Long's test $p > 0.05$, Fig 5B). This provides further evidence that

Pax7 target gene repression is at least as major a signature as DUX4 target gene expression in FSHD skeletal muscle.'

Reviewer Comment: As the work features many qPCR experiments, it would be ideal if the Authors could follow the MIQE standards to improve the power of their findings. Methodology is not detailed. What is the normalizer? How did the authors choose it? What kind of statistical test was done to choose the normalizer? If the normalizer is not chosen properly, all the results are subject to change.

RESPONSE: Shen et al. (2010) previously demonstrated that the gene RPLPO is a reliable gene to normalise gene expression in proliferation and throughout differentiation. In line with this and other publication in the field we have selected RPLPO as stable reference gene our study. A more detailed methodology is now included into materials and methods. MIQE guidelines for RT-qPCR were followed.

Reviewer Comment: In figure 2, how was the statistical analysis performed? Did they compare FSHD myoblasts and control myoblasts on one hand and FSHD myotubes and control myotubes on the other hand or were the two cell states in fact combined (as is suggested by bars indicating statistical significance)? What happens if a MANOVA statistical analysis is done?

RESPONSE: The original Figure 2 which described the comparison of our Pax7 target gene score in control cell line 54-6 to FSHD cell line 54-12 has been replaced by Figure 5, in which we now compare 5 FSHD lines to 4 control lines (3 patients and 3 matched controls, see above for details). Only confluent myoblasts are now presented. Please see the response to the above point for a description of statistics employed.

Reviewer Comment: In figure 3, did the authors compare Pax7 transfected cells versus all the other conditions at the same time? The experimental method is poorly described. What is p34 exactly? What is the DUX4 expression plasmid used? What quantity of plasmids was used for the transfection? A T-test is not appropriate in this figure.

RESPONSE: The methods has been amended as follows, t-tests were employed post hoc after ANOVA revealed significant differences between groups.

'HEK-293 and NIH-3T3 cells (12500 cells/well) were co-transfected with plasmids encoding DUX4, Pax7, dominant negative DUX4-ERD, dominant negative Pax7-ERD, or DUX4 and Pax7 all encoded in pMSCV-IRES-eGFP, together with reporter plasmids encoding either DUX4-responsive promoters driving luciferase expression (pZscan4-luc34 or RFPL4b-luc) or a Pax3/7-responsive element driving β -galactosidase (p34 plasmid) using Lipofectamin LTX (ThermoFisher). A pRSV vector encoding either β -galactosidase or luciferase as appropriate was co-transfected as an internal control for normalization of transfection efficiency. Total amount of DNA transfected per reaction was 1.5 μ g, in equal ratios for reporters and plasmids encoding genes of interest, and 250 ng/reaction of the internal normaliser vector. Cells were harvested 24 h after transfection, and assayed using the Dual-light Reporter system (ThermoFisher) according to manufacturer's instructions. Three independent transfections were performed in technical triplicate. Reporter gene activity was measured using Glomax-Multi+ plate reader (Promega) and normalised for transfection efficiency.

N=3 for each cell line, ANOVA revealed significant intensity differences, post hoc two tailed t-tests were employed to assess significant pairwise differences.'

Reviewer Comment: Why did the authors chose to transfect a plasmid expressing murine Pax7 instead of Human PAX7?

RESPONSE: To derive a set of PAX7 targets we assayed primary murine satellite cell-derived myoblasts over-expressing Pax7, a dominant negative version Pax7-ERD and control retroviral constructs. We assayed gene expression via microarray to facilitate compatibility with the publically available FSHD muscle biopsy data sets. Primary murine satellite cells were selected as current evidence suggests co-expression of PAX7 and DUX4 in a muscle progenitor (satellite cell) phase during FSHD myogenesis, and widespread epigenetic changes have been shown to occur subsequently to this stage, which may alter transcription factor binding dynamics. Human satellite cells are less readily available to us, hence we used murine cells and murine Pax7. PAX7 is highly conserved across species and it is likely that targets derived from this murine setting will be representative of over-expression of PAX7 in human primary satellite cells.

Reviewer 1 comments in their second reviewer response: 'Pax7 plays a major role in both Human and murine muscle physiology and a conserved set of activated genes is expected. Reviewers 3 also comments: 'The species differences seem to be less of a concern with the PAX3/7 datasets since these genes are well-conserved in mammals, and even further back in lower organisms, such as zebrafish.'

Hence we feel that the murine setting is sufficient to generate a reliable set of Pax7 target genes which are representative of human targets.

Reviewer Comment: The results shown in Figure 3A are intriguing: Pax7 over-expression induces p34 reporter activity. Similar results are obtained when DUX4 is over-expressed. However, when both DUX4 and Pax7 are expressed, no change in reporter activity is observed. This needs to be discussed. What could be the possibilities having in mind the homologies between the DNA binding domains of DUX4 and PAX3/7?

RESPONSE: Indeed we find this intriguing as well, however, as Reviewer 3 comments: 'Cotransfection experiments can be subject to artifact related to efficiency of DNA uptake' hence to further investigate this we also measured transcript expression of validated Pax7 target SELP during co-transfection (and for the converse measured the expression of 4 validated DUX4 targets). We found that while SELP was induced by Pax7 (in a manner which was mitigated by co-expression with DUX4), it was not induced by DUX4 like the p34 reporter. It is possible that the interactions between DUX4 and Pax7 are not limited to competitive DNA-binding and that there may be some protein level interaction leading to suppression of transcriptional targets below the levels expected by competitive binding but the data on this phenomenon is unclear. What is clear is that co-transfection with Pax7 and DUX4 in the same cell results in suppression of Pax7 transcriptional activity relative to Pax7 expression alone, as we comment in the manuscript.

Reviewer Comment: In figure 3B, the co-transfection of DUX4 and Pax7 does not activate the DUX4 reporter construct. But in Fig 3C, it does. How do the authors explain this difference?

RESPONSE: The two DUX4 reporters considered are based on sequences from different genes, and so it is unsurprising that they do not behave identically. While Pax7 activates the RFPL4b reporter, it is to significantly lower levels than DUX4, and co-transfection of DUX4 with Pax7 mitigates this activation. To further investigate, we also performed RT-qPCR of native *ZSCAN4* and *RFPL4b*, as well the validated DUX4 targets *TRIM48* and *MBD3L2*. We found that at the transcript level, Pax7 is unable to activate these targets while DUX4 was. Importantly, co-transfection of Pax7 and DUX4 acted to reduce the transcript levels of all 4 DUX4 target genes below the levels achieved with DUX4 alone. Hence while Pax7 may induce some activation of the RFPL4b-based DUX4 reporter construct, it does not change transcript levels of the native gene.

Reviewer Comment: The conclusion of the authors concerning these experiments is that co-expression of Pax7 and DUX4 acts to suppress the transcriptional targets of both DUX4 and Pax7, which is obviously not the case (fig 3C).

RESPONSE: The conclusion that we intended to highlight was that DUX4 and Pax7 together suppress the respective reporters relative to expression of either DUX4 or Pax7 alone. We hope that this is now clear in the revised manuscript.

Reviewer Comment: The GSEA analysis is of interest but the authors could examine more closely the genes involved, for example using Leading Edge Analysis (Leading Edge genes are those that drive the up- or down-regulation of the gene sets - technically they are identified by GSEA to lie before the maximum of the running enrichment score). For example, it is curious that only one Hypoxia-related gene set out of 48 was significantly enriched - which genes drive this enrichment? Are they a subset of genes that are special to that hypoxia-related gene set that are not found in other hypoxia-related gene sets? How do these hypoxia-related leading edge genes compare to those hypoxia-related genes that the authors identified to be dysregulated in their previous work? As a second example, it is very interesting that Hypoxia and EZH2 targets are both enriched in both PAX3 and PAX7 - do the leading edge genes of these two gene sets overlap with each other? Do they overlap between the PAX3 and PAX7 studies?

The two examples relate to a more basic question that the authors could address: is there a small subset of genes that are strongly regulated by PAX3 and PAX7 (a subset that overlaps with Hypoxia and with EZH2 targets)? It would also be of interest to see the GSEA enrichment curves for these two gene sets for PAX3 and PAX7.

RESPONSE: The PAX3 target gene repression angle has now been omitted from the paper due to reviewer concerns over the use of forkhead fusion proteins. The GSEA results are thus limited to the gene sets associated with up and down regulated Pax7 targets. We appreciate the reviewer's points regarding leading edge analysis of the GSEA results, these would indeed provide potentially interesting information. However we did not perform GSEA using the Kolmogorov Smirnov test which permits the generation of enrichment curves and performance of leading edge analysis. Instead we performed GSEA using a Fisher's Exact test evaluating the unweighted overlap of our Pax7 targets with the genes in Molecular Signatures Database (as detailed in the Methods). The reason we performed this particular analysis was because it is unclear which data one should use to weight the Pax7 target genes for Kolmogorov-Smirnov based GSEA, several options present which will likely give different results: (1) Differential expression levels of genes under Pax7 over-expression vs control. (2) Differential expression levels of genes under Pax7-ERD over-expression vs control. (3) Differential expression levels of genes in FSHD muscle biopsy samples vs controls (in this case there are further questions: which of the 6 biopsy data sets is most appropriate? Do we use a summary statistic for each data set? If so how do we deal with the differences between RNA-seq and microarray studies?). (4) Differential expression levels of genes in FSHD myoblast cell lines vs controls. Thus, rather than admit the ambiguity of choosing a weighting, we decided instead to perform an unweighted form of GSEA using the Fisher's Exact test.

We provide full tables of the top 100 GSEA results for both Pax7 activated and suppressed targets as supplementary tables in the new version of the manuscript, where the reviewer can find precisely which genes are associated with each enriched gene set.

Reviewer Comment: In their conclusions, the authors declare that DUX4 expression inhibits both proliferation and myogenic differentiation, a phenotype similar to that observed when PAX3/7 expression is repressed. However, in the Krom et al article describing the

immortalized clones, the fusion index was improved in the FSHD clones. How do the authors explain that?

RESPONSE: We have removed this tentative discussion point from the updated manuscript, as we have focused on myoblasts rather than myotubes for the validation of the Pax7 and DUX4 target gene signatures.

However, the reviewer raises an interesting point regarding the Krom et al. mosaic cells: We have robustly characterised several lines used in this paper and in particular we note that whilst the pathological clones described by Krom et al. may display an increased fusion index, this seems to be attributable to an attrition of cells outside of myotubes rather than an increased fusion rate itself. In fact the pathological clone we considered in the previous submission displays an atrophic myotube morphology, characterised by reduced MyHC+ area on immunostaining.

FSHD cell line 54-12 fuses to form atrophic myotubes.

(A) Immunofluorescence staining of MyHC shows atrophic myotube formation of FSHD myoblast cell line 54-12 compared to control cell line 54-6 at day 2, day 3 and day 5 of differentiation. (B) Quantification of myotube size by MyHC+ area shows that at day 2, day 3 and day 5 of differentiation, FSHD cell line 54-12 displays smaller myotubes than control cell line 54-6. Cells were assayed in triplicate, an asterisk denotes the significance of a two tailed t-test ($p < 0.05$).

Reviewer Comment: Finally, considering that (1) most of the articles using primary FSHD cells have shown that DUX4 expression is barely expressed in FSHD myoblasts whereas it is highly expressed in FSHD myotubes and (2) satellite cells account for 2-7% in adult muscles, and (3) PAX7 is only expressed in satellite cells, how can it be that suppression of PAX3/7 transcriptional targets is the hallmark of FSHD? The discussion about low and high

levels of DUX4 in muscle fibers is interesting and should be more developed to incorporate these points. Where could be the most important role of DUX4 in the physiopathology of FSHD: in the muscle fiber (leading to its apoptosis) or in the satellite cells (inhibiting myoblast differentiation)? Is there any link between the number of satellite cells per fiber and FSHD affected muscles? All important points, discussion!

RESPONSE: We agree that these are all interesting points for discussion. To address point (1) Studies have shown high levels of DUX4 in myotubes, and lower levels in myoblasts. However, we believe that the interaction between DUX4 and Pax7 likely occurs before the myoblast stage in the satellite cell phase where Pax7 level are high. A recent study by Caron et al. 2016, showed elevated DUX4 expression in such a Pax7+ cell population differentiated from human ES cells from FSHD patients, which decreases once the cells differentiate to myoblasts and rises again when myotubes are formed. Moreover we have recently demonstrated in Knopp et al. 2016 that DUX4 expression increases during regeneration of an FSHD mouse model, concordant with co-expression of DUX4 and Pax7 in a satellite cell phase. For points (2 & 3) it is indeed interesting that Pax7 target gene repression is a detectable signal in muscle biopsies despite the low level of satellite cells in such biopsies. We have shown that suppression of Pax7 target genes is not a feature of Duchenne muscular dystrophy where satellite cell numbers are reduced so it is unlikely that this feature of FSHD skeletal muscle is driven only by the satellite cell pool. Rather we believe this suggests that transcriptional changes brought about in the satellite cell stage may persist following differentiation. Such a phenomenon has been previously described with Pax7 inducing wide-spreading de-methylation resulting in persistence of transcriptional changes (Carro et al. 2016). Moreover we here have identified that Pax7 perturbs epigenetic modifiers such as EZH2, which may further promote persistence of transcriptomic changes.

In the revised manuscript, we report new analysis and new data that supports DUX4 target gene expression FSHD muscle biopsies. However Pax7 target gene repression is still a hallmark of FSHD skeletal muscle and our data demonstrates that it is at least as major as DUX4 target gene expression. This has led to an updated hypothesis in which DUX4 targets and Pax7 targets are at least equally important contributors to the FSHD molecular landscape; which is dominant is unclear at this stage and we do not wish to speculate on current evidence. However in discussion we propose that both mechanisms merit investigation and as the reviewer will be aware, the investigation into the latter mechanism is currently much more limited.

Response to Reviewer 2

First Round comments

Reviewer Comment: Zammit and colleagues address an important question in the field of FSHD, namely the lack of consistency in transcriptional profiling data in cells and tissue samples from FSHD patients. Although the DUX4 gene is known to cause FSHD, the mechanism by which it acts to cause muscle pathology is unknown. At high levels of expression, DUX4 induces a group of genes, collectively described as the DUX4 signature. Zammit and colleagues look for this signature in a large set of published data and do not find it. This is a very provocative and important result.

Rather, they find an inverse PAX7 signature, meaning that genes that are regulated by Pax7 are found to be inversely regulated in FSHD samples. The authors speculate that this may be an effect of low level DUX4 expression, distinct from the DUX4 signature seen with high level expression, in which DUX4 antagonizes PAX7. The connection to PAX7 is quite interesting as the two proteins have similar homeodomains and it has been shown that PAX7 and DUX4 can compete with one another.

This study is an important contribution to the field and will be received with great interest. There are however a few issues that should be addressed:

1. The authors evaluate expression in immortalized isogenic myoblast cell lines. Although the original study describing these lines reported derivation of multiple FSHD and control clones from a mosaic individual, and indeed drew conclusions from evaluating multiple clones, the authors only use a single clone. As the previous paper showed significant inter-clonal variation, data from comparing two groups of n=1 clone is not reliable. The authors should repeat this part of the study using the full panel of isogenic clones.

RESPONSE: A good point, as mentioned above in the response to Reviewer 1: we now compare 5 FSHD lines to 4 control lines. To confirm the results hold despite inter-clonal variability in the Krom et al. mosaic lines we consider 3 pathological (54-12, 54-2 and 54-A5) and 2 control lines (54-10 and 54-6), as well as 2 additional pathological lines 12A and 16A from the Wellstone centre and matched controls 12U and 16U.

Reviewer Comment: 2. In Figure 2, the DUX4, PAX3 and PAX7 scores are shown for myotubes and myoblasts for FSHD and control lines. Shouldn't myotubes have a lower PAX3 and PAX7 score than myoblasts? They ought to have no or low levels of Pax7. Yet the levels are similar. Please explain.

RESPONSE: In the revised manuscript, we only consider cultured myoblasts from multiple patients to simplify the interpretation.

We note that in the original submission these scores were z-normalised within each condition so are not comparable and that they correspond to Pax7 target gene expression rather than Pax7 expression, recent studies have shown that Pax7 can induce widespread de-methylation which results in persistence of transcriptional targets Carrio et al. 2016.

Reviewer Comment: 3. A further concern about the PAX3/7 signatures is that they may be detecting "quality" of myoblasts. Perhaps myoblast cultures from diseased muscle simply show a reduced score on this metric, perhaps because they tend to be more prone to differentiate or are more prone to fibroblastic contamination. To address this, please evaluate the PAX3/7 signature on independent datasets, specifically diseased vs. control myoblasts from published Duchenne MD profiling studies.

RESPONSE: This is a good point, we have now shown that meta-analysis across 4

Duchenne MD muscle biopsy studies reveals no suppression of Pax7 target genes in Duchenne (Supplemental figure S3). This confirms that the suppression is FSHD specific and not a consequence of myoblast 'quality'.

Reviewer Comment: 4. Cotransfection experiments can be subject to artifact related to efficiency of DNA uptake. Can Figure 3 be supported with data on endogenous genes, in addition to the transfected reporters? I.e., in addition to looking at the co-transfected ZSCAN4 reporter, what is the level of the endogenous ZSCAN4 gene? Of some endogenous Pax7 target genes.

RESPONSE: This is a good point and is discussed above in response to Reviewer 1. We have evaluated transcript levels of the Pax7 target gene SELP and 4 DUX4 transcriptional targets ZSCAN4, RFPL4B, TRIM48 and MBD3L2, which confirm the findings of the reporter assays.

Reviewer Comment: 5. The authors only evaluate DUX4 and PAX3/7 signatures. It would be valuable to test some negative control signatures. How significant is the inverse Pax7 signal vis a vis other potential signals? I.e. if you selected lineage-specific transcription factors at random, would they show correlations in the FSHD data.

RESPONSE: As a negative control we have performed a resampling procedure generating 1000 random target gene sets of equivalent size to Pax7 target gene sets and re-running meta-analysis across FSHD muscle biopsy microarray data sets. We never observed any values close to those seen for PAX7, thus a conservative estimate for the probability of observing the PAX7 target gene repression by chance is $<10^{-4}$. This is presented at Supplementary Figure S5 in the revised manuscript.

Reviewer Comment: 6. Along the same lines, it has been shown that DUX4 has effects on MYOD. Can the authors evaluate the MYOD signature to determine whether FSHD vs control samples show alterations in this?

RESPONSE: To evaluate MYOD targets we utilised the DELASERNA_MYOD_TARGETS_UP and DELASERNA_MYOD_TARGETS_DOWN gene sets available on the Molecular Signatures Database published by de la Serna et al. 2008, to construct a MYOD target gene score analogous to our Pax7 target gene score. We found that although no individual data set was significant, there is a trend towards repression of MYOD across microarray studies of FSHD muscle biopsies (in line with the findings by ourselves and others that DUX4 represses MYOD), but this was also not significant on meta-analysis. We include this result as Supplementary Figure S1.

Response to Reviewer 3

Reviewer 3: response to our initial reply to reviewers:

Reviewer Comment: I am still concerned about the groups they used to generate their gene expression signatures, and it does not appear that those will change'

RESPONSE: We appreciate this point and have changed the data sets used to generate the gene expression signatures as discussed in above in response to Reviewers 1 and 2. We have removed all murine DUX4 over-expression data sets and removed all datasets considering forkhead fusion genes.

First Round Comments

Reviewer Comment: A. Summary: The DUX4 gene, linked to FSHD, is a transcription factor. A number of mechanisms by which DUX4 could contribute to FSH muscular dystrophy have been proposed and are under investigation in the field. Most of these mechanisms involve DUX4 activation of target genes that have been implicated in a number of potentially deleterious pathways. An alternative mechanism, proposed several years ago by Bosnakovski, et al (reference 5 in the current manuscript) suggests that DUX4 acts by interfering with PAX3/PAX7-derived gene expression, and that PAX3/7 target gene repression may play a role in FSHD. The current manuscript by Banerji, et al explores this hypothesis. To do this, they perform meta analysis of several published gene expression studies in which DUX4/DUX4-fusion proteins or PAX3/7 or PAX3/7-forkhead fusion proteins were over-expressed in various cells. They also performed an additional PAX7 / PAX7-fusion protein over-expression study themselves, to provide some new data to the analysis. They used these over-expression studies to generate DUX4-, PAX3- and PAX7-expression signatures, and then queried previously published gene expression datasets derived from FSHD cells/biopsies in which DUX4 was present (or presumably present, since it wasn't measured in many cases) at extremely low, endogenous levels. Based on these analyses, the authors conclude that DUX4 target genes are not changed in FSHD muscles, but that PAX3/7 target genes were significantly repressed, lending support to the hypothesis that inhibition of PAX3/7 target genes are a contributing factor to FSHD pathogenesis.

RESPONSE: Following a more extensive analysis using data derived from human cells, published after our first submission, we were able to demonstrate a weak but significant upregulation of DUX4 targets on meta-analysis of FSHD muscle biopsies. Our updated conclusions are that Pax7 target inhibition is as major a molecular signature as DUX4 target gene expression.

Reviewer Comment: B. Originality and interest: From the standpoint of this study being a meta-analysis largely utilizing others' published data, a large amount of the actual data is not new. However, there have been a number of FSHD gene expression studies performed, as noted by the authors, and in studies that did not utilize DUX4 over-expression methods, there seemed to be little overlap in expression changes among the various published works. Nevertheless, a rigorous statistical comparison has not been published, to my knowledge. So an analysis like the one here is long overdue, and as such, a very welcome and important pursuit for the field.

RESPONSE: We thank the reviewer for their positive comments on our statistical analysis and note that in the updated submission we have contributed a larger new dataset of RNA-sequencing of immortalised human myoblasts from FSHD and control individuals to confirm our findings. We hope this will improve the perceived originality of our work.

Reviewer Comment: C-F. The authors defined their DUX4, PAX3 and PAX7 expression signatures using datasets in which each respective protein was overexpressed. On this surface, this seems logical and sensible. However, there are a few issues with this approach that need consideration and could detract from the power of this analysis.

Issue 1: Differences in cell type and species used to generate gene expression signatures.

Specifically, the authors used three different datasets to generate their DUX4 data. Two were performed in two different kinds of mouse cells (mouse primary satellite cells, and mouse C2C12 myoblasts), while the third was performed in immortalized human myoblasts. These differences could have a large influence on gene expression changes, thereby calling into question the reproducibility of the data from one experiment to the next. Specifically, close DUX4 homologs are present only in primates. Mice do not contain DUX4, although they do express a paralogous gene (Dux) that is structurally similar to DUX4 but fairly divergent on the amino acid level. Certainly there are a number of studies supporting that mouse can be a useful model to study DUX4 activity, and this reviewer believes that the mouse has utility for studying various aspects of DUX4 and FSHD. However, considering the two genes have been under different selective pressures for millions of years, and there are no published data suggesting they have maintained the same gene targets, it is not clear to me that a DUX4 gene expression profile generated in mouse cells is comparable to one obtained in human cells. Thus, the reproducibility of the DUX4 over-expression datasets used to generate DUX4 gene signatures comes into question. The species differences seem to be less of a concern with the PAX3/7 datasets since these genes are well-conserved in mammals, and even further back in lower organisms, such as zebrafish.

RESPONSE: With new data recently published, we are now able to consider human myoblasts over-expressing DUX4 to derive our signatures, following methodologies previously outlined by others to permit easy comparison. To avoid reiteration, please see above, the detailed response in relation to a similar point raised by Reviewer 1.

Reviewer Comment: Issue 2: A second issue related to the over-expression datasets for generating gene-specific expression signatures is the use of fusion proteins. Specifically, the PAX3 and PAX7 datasets relied upon PAX3-forkhead, and PAX7-forkhead proteins derived from translocations found in rhabdomyosarcomas. Thus, the transactivation domains are different from the natural PAX3 and PAX7 proteins. Is this the same as PAX7 and PAX3? Is it legitimate to compare PAX3 and PAX7-derived datasets to ones utilizing these translocations (which are different proteins)? Moreover, one of the DUX4 datasets, and one of the PAX7 datasets, also used fusions to an ERD repressor domain, or in the case of DUX4, to a VP16 transactivation domain. Considering these C-terminal regions may contain binding sites for transcriptional co-factors, and they are different in many cases, is it possible that these different transactivation regions could influence gene expression profiles, thereby making the determination of a PAX3-, PAX7-, or DUX4-specific expression profile less reliable? I am uncertain about this.

RESPONSE: We appreciate this point regarding the fusion proteins which is also raised by Reviewer 2 and responded to above. We have omitted all signatures derived from consideration of forkhead fusion proteins and also omitted the DUX4 signature derived from the murine data set over-expression the DUX4 VP16 fusion gene, which as the reviewer points out may mask the importance of the C-terminus. We note that as a consequence of removing the forkhead fusion data sets we have removed the analysis of the Pax3 signature and focused only the Pax7 and DUX4 signatures.

Reviewer Comment: Issue 3: Performing a meta-analysis like the one here obviously relies upon others' published work. In this case, the authors of this manuscript compared 5 gene expression studies in which biological materials were analyzed using different expression

arrays, from at least three different muscle groups, in different populations of FSHD patients. It was unclear, however, how many of these materials showed DUX4 expression, and if so, at what levels? Do the authors of this manuscript have this information? If so, it should be included in Suppl Table 1. Moreover, would the authors expect that the presence/absence of DUX4 would impact the presence/absence of a DUX4-target gene expression profile in a dataset? Likewise, would the presence/absence of DUX4 impact its ability to interfere with PAX3/7 genes?

RESPONSE: We have included the reported DUX4 expression levels in FSHD muscle biopsies in the Supplementary Table S6 describing the datasets. Only 1/6 data sets report DUX4 expression in FSHD biopsies, 4/6 data sets report no DUX4 expression, and DUX4 levels were not reported in 1/6 data sets. We have provided a discussion as to how a transient expression of DUX4 may interfere with Pax7 target gene expression during regeneration leading to long term transcriptional changes in the updated manuscript. Hence it is possible that Pax7 target gene expression may be a consequence of DUX4 expression even if DUX4 itself is not detectable in the muscle biopsies.

Reviewer Comment: Other comments related to the data and data presentation: as this type of analysis has not been performed before, I think it would be extremely useful to include a summary of each study used in this manuscript. Specifically, how many genes were on each microarray? How many total genes were significantly changed (up or down)? How many were in common between each individual study used here? How many differences?

RESPONSE: We appreciate that a comparison of differential expression identified in FSHD muscle biopsies is important. However, it would be inappropriate to simply report the number of differentially expressed genes reported in the publication of each biopsy dataset utilised, as different investigators employed different statistical methodologies so these numbers are not comparable. We have performed a more standardised analysis of differential expression across multiple of the FSHD muscle biopsy data sets and reported results in Banerji et al. 2015, where the reviewer can find the requested information.

Reviewer Comment: It would also be useful to have the same analysis done with a control transcription factor (preferably a homeodomain containing protein) that is unlinked to the FSHD pathogenic mechanism. Are similar gene expression data available for other HOX genes?

RESPONSE: Unfortunately there are not similar gene expression data available for other HOX genes over-expressed in muscle, and given concerns over some of the data sets we previously used in our previous analysis, we feel that it would be inappropriate to examine over-expression of HOX genes in different cell lines. As a negative control, we have performed a resampling procedure to simulate the discriminatory power of 1000 random gene sets of equivocal size to the Pax7 target gene set as a control. We find that no random gene set is able to display the discriminatory power of the Pax7 target gene repression signature across FSHD microarray muscle biopsy datasets, presented as Supplementary Figure S5 in the revised manuscript.

Reviewer Comment: On lines 51-53 of the manuscript, the authors stated that "A single sample score for each transcription factor was constructed via comparison of activated and repressed target gene expression levels in a given sample, with genes in multiple transcription factor scores removed to ensure independence". Could the authors please clarify this methodology and rationale for doing so in greater detail, perhaps also by providing an example of how removing such genes impacts the score? How many genes were excluded? Is it worthwhile to include a list of excluded genes in the supplement?

RESPONSE: This is a good point. We understand that this score construction may introduce unnecessary ambiguity and so we now construct our DUX4 and Pax7 target gene scores without removing the overlapping genes. We include tables describing the target genes used to construct each score (Supplementary Tables 1-3).

Reviewer Comment: The data in Figure 2, using the mosaic patient cells, are interesting, potentially important and seem to support the authors' meta analysis. The major drawback to these data are that they are derived from N=1 patient.

RESPONSE: As discussed in response to similar points made by Reviewers 1 and 2, we now include RNA-seq data from 5 FSHD lines corresponding to 3 independent patients and 4 matched control lines (corresponding to 3 individuals). The new RNA-seq data confirms our findings that Pax7 target gene repression is as major a molecular signature of FSHD myoblasts as DUX4 target gene expression.

Reviewer Comment: Finally, I'm uncertain that the title accurately reflects the paper. There are two issues related to my way of thinking here. First, considering the concerns laid out above (related to differences in cell type, species, and wild-type versus fusion proteins) is it completely accurate to lump the expression changes into bins of PAX3-, PAX7- and DUX4-target genes as stated in the title and throughout the manuscript? Second, if we put this argument aside, and focus solely on the data as presented, the authors show that DUX4 transcriptional targets are not observed, while PAX3/7 target genes are reduced. Considering this only, one could argue then that the title does reflect the data, but not the conclusions. Specifically, in the conclusions, the authors suggest that DUX4 likely operates by inhibiting PAX3/7 genes, and although their data do not show it, they suggest DUX4 could still contribute to FSHD pathogenesis by activating its own targets.

RESPONSE: We agree on the limitations of the original title and have modified the title of the revised manuscript to 'PAX7 target genes are globally repressed in FSHD skeletal muscle', we believe this title more accurately reflects the results and conclusions of our manuscript.

Reviewers' comments:

Reviewer #1 (Remarks to the Author):

The article presented by Banerji et al has been greatly improved. The authors have modulated their message and their new analysis now demonstrates that both PAX7 and DUX4 target gene expression are biomarkers of FSHD.

In particular, DUX4 over expression experiments in murine cells have been removed from the meta-analysis and most of the important points raised in the review have been now addressed. However, the biomarker of Pax7 expression is determined in murine cells and then evaluated human FSHD biopsies.

In Fig6A/B, how do the authors explain that DUX4 alone increases P34 reporter activity/SELP transcript levels? The explanation given by the authors is that DUX4 can bind and activate the Pax7 reporter genes, but when expressed with Pax7, its effect is to repress activation of Pax7 transcriptional targets. What could be the biological explanation?

On simple explanation for loss of Pax7 expression could be a loss of myoblasts/satellite cells in FSHD muscle biopsies. Have the authors check this hypothesis? Is it possible to normalize the biomarker of Pax7 expression with a gene only expressed in satellite cells? It would have been interesting to perform a PAX7 staining on primary FSHD and control myoblasts.

Finally, an article has been very recently published showing that Expression patterns of FSHD-causing DUX4 and myogenic transcription factors PAX3 and PAX7 are spatially distinct in differentiating human stem cell cultures (Haynes 2017). How do the authors conciliate this article and their data?

Minor correction: lines 234 Fig4A should be Fig4A-D, and line 240 Fig4B should be Fig4E.

Reviewer #2 (Remarks to the Author):

The authors have performed an extensive set of new experiments to address the concerns of the three reviewers, and their revised manuscript is now significantly improved. The central message stands intact as well.

This will be a controversial study because much of the field is invested in a particular model for how DUX4 causes muscle pathology, and the data here cause one to rethink this model. The pattern that the authors have discovered is clear and presumably will be subject to replication by others and in new profiling data sets as they emerge, so it seems fair and appropriate to me to give this data a platform. I see nothing fundamentally wrong in the approach or the methodology. My own criticisms addressed various non-specific ways in which the authors might be seeing such a pattern and the authors have tested these and eliminated them, therefore as controversial as these results will be, they deserve to be given a platform.

Minor point: regarding DUX4 repression of MyoD, the authors reference their own paper from 2016. The original demonstration that DUX4 represses MyoD was published in 2008: Bosnakovski et al., EMBO J.

Reviewer #3 (Remarks to the Author):

My major concerns about the original submission were centered on the methodology, which relied

upon comparing datasets between species (mouse and human) as well data obtained from expressing various fusion proteins.

The authors have removed those datasets and added some additional human myoblast data. Doing so has changed the results and conclusions of the paper somewhat, but I think the overall message is maintained: interference with PAX7 target genes may be a signature of DUX4 expression. They now also report finding some evidence for DUX4 target gene activation as well.

In my opinion, this paper will receive important consideration from the FSHD research field.

Response to Reviewers

Reviewer #1 (Remarks to the Author):

Reviewer 1 comments: *The article presented by Banerji et al has been greatly improved. The authors have modulated their message and their new analysis now demonstrates that both PAX7 and DUX4 target gene expression are biomarkers of FSHD.*

In particular, DUX4 over expression experiments in murine cells have been removed from the meta-analysis and most of the important points raised in the review have been now addressed.

RESPONSE: We are pleased that Reviewer 1 agrees that most of their ‘important points’ have been adequately addressed in the first revision.

Reviewer 1 comments: *However, the biomarker of Pax7 expression is determined in murine cells and then evaluated human FSHD biopsies.*

RESPONSE: PAX7 is highly conserved across species, with 97% amino acid sequence homology overall between mouse and man. Indeed, Reviewer 1 actually stated in their first set of comments that “Pax7 plays a major role in both Human and murine muscle physiology and a conserved set of activated genes is expected.” Reviewer 3 also noted in their previous review that: “The species differences seem to be less of a concern with the PAX3/7 datasets since these genes are well-conserved in mammals, and even further back in lower organisms, such as zebrafish.”

We have amended the re-revised manuscript to more clearly emphasize the conservation in protein sequence between mouse and man.

Reviewer 1 comments: *In Fig6A/B, how do the authors explain that DUX4 alone increases P34 reporter activity/SELP transcript levels? The explanation given by the authors is that DUX4 can bind and activate the Pax7 reporter genes, but when expressed with Pax7, its effect is to repress activation of Pax7 transcriptional targets. What could be the biological explanation?*

RESPONSE: DUX4 and Pax7 are both able to activate the p34 Pax7 reporter and induce expression of the Pax7 transcriptional target gene SELP, but the effect, as expected, is greater with Pax7. Such DUX4-mediated activation of Pax7 target genes is likely due to DUX4 binding to the Pax7 binding site due to homology between the homeodomains of the two proteins (Bosnakovski et al. (2008) *EMBO J.* **27**, 2766-79). Importantly though, when DUX4 and Pax7 are co-expressed, we observe lower activation of p34 and SELP than with expression of either Pax7 or DUX4 alone. Thus, this is consistent with our model of DUX4 and Pax7 being able to mutually suppress the ability of the other to activate its target genes. For example, Pax7 may preferably bind/have higher affinity for the sites in p34/SELP, but DUX4 may sequester away a co-factor necessary for Pax7 to activate target genes, or Pax7/DUX4 proteins may physically interact resulting in suppressed gene activation.

We have now amended the results and discussion of the re-revised manuscript to expand on the potential mechanisms by which PAX7 and DUX4 mutually suppress transcriptional activity.

Reviewer 1 comments: *On simple explanation for loss of Pax7 expression could be a loss of myoblasts/satellite cells in FSHD muscle biopsies. Have the authors check this hypothesis?*

RESPONSE: It has recently been reported that there is no significant difference in satellite cell numbers in FSHD versus control muscle biopsies (Statland et al. (2015) *J Neuromuscul Dis* **2**, 291-99).

In addition, this point was initially raised by Reviewer 3 in the first round of review, who suggested that we evaluate our Pax7 signature on Duchenne muscular dystrophy biopsies, where a loss of myoblasts/satellite cells is widely reported. We performed such analysis and found no significant difference in PAX7 target gene expression in Duchenne vs control muscle biopsies (included in the original revision as Supplementary Figure 3). This indicates that suppression of PAX7 target genes is not a universal consequence of muscular dystrophy but is specific to FSHD in a manner not driven by loss of myoblasts/satellite cells *per se*.

We have augmented the relevant section in the re-revised manuscript to better explain these implications.

Reviewer 1 comments: *Is it possible to normalize the biomarker of Pax7 expression with a gene only expressed in satellite cells?*

RESPONSE: The Pax7 target gene set was derived from satellite cells, and so our Pax7 signature will inherently normalise out some satellite cell contribution. Furthermore, it is not apparent which gene to use that is specific to satellite cells but would not be present in myoblasts, and/or other cell types in a biopsy of whole muscle tissue: any choice will bias the biomarker to variations in expression of that gene, potentially masking the signal from PAX7 targets. The contribution of the satellite cell pool to a whole muscle biopsy is small, hence any satellite cell specific marker will be expressed at very low levels and thus subject to greater noise. Consequently normalisation to a satellite cell specific gene will introduce greater uncertainty to our results. Importantly, as we are evaluating our Pax7 biomarker on FSHD muscle biopsies and immortalised proliferating FSHD myoblast cultures (not satellite cells), normalising to satellite cells cannot be performed for both. Finally, as discussed above, there is no difference in satellite cell numbers in FSHD and control muscle biopsies, so normalising to account for satellite cell abundance will have minimal impact.

We hypothesize that Pax7 target gene repression in FSHD is not driven by loss of satellite cells, but is rather a consequence of DUX4 disruption of Pax7-generated transcriptional landscape at a muscle progenitor stage that persists in mature muscle through Pax7-induced epigenetic changes. Thus normalising our signature to a satellite cell marker would also mask the true signal.

Reviewer 1 comments: *It would have been interesting to perform a PAX7 staining on primary FSHD and control myoblasts.*

RESPONSE: We are investigating expression of PAX7 target genes in FSHD, rather than expression of PAX7 itself, so feel that this would distract from the main focus of the manuscript. We have performed immunostaining and RT-qPCR analysis for PAX7 in immortalised FSHD1 (54-12) and control (54-6) myoblasts isolated from a mosaic patient and found that both clones showed comparable Pax7 mRNA levels and nuclear localised PAX7 protein.

Reviewer 1 comments: *Finally, an article has been very recently published showing that*

Expression patterns of FSHD-causing DUX4 and myogenic transcription factors PAX3 and PAX7 are spatially distinct in differentiating human stem cell cultures (Haynes 2017). How do the authors conciliate this article and their data?

RESPONSE: Haynes et al. (2017, *Skeletal Muscle* 7, 13) examined DUX4, PAX7 and PAX3 expression dynamics in FSHD myoblasts/myotubes differentiated from the embryonic stem cell state using human ES and iPS cells isolated from FSHD patients and controls. PAX3 or PAX7 protein is readily detected in cells at D7, D21 and D30 but no DUX4 protein containing cells were found (Table 2), despite PAX7 and DUX4 message being simultaneously highly expressed at the myogenic progenitor stage D21 (Figure 3). Even at the myotube stage (D40), only 3/4909 cells in one line and 43/4460 cells in another were reported to contain DUX4 protein, while it was 70 and 1678 cells respectively with PAX7 protein (Table 2). Apart from the potential co-localisation of PAX7 and DUX4 at any of the many stages not assayed by immunolabelling (e.g. between D0-6, D8-20, D22-29 and D31-39), an obvious caveat is that DUX4 may kill any PAX7-containing cell with higher (detectable) levels of DUX4 before the myotube stage. Indeed, the number of co-expression events between PAX7 and DUX4 detected is significantly less than would be expected, and the authors themselves comment on the possibility of mutual exclusivity of the two proteins: “**An additional caveat is the possibility that DUX4 expression reduces expression of PAX3 or PAX7 making their expression mutually exclusive**”. Such mutual exclusivity would ensure that a DUX4 expressing population would naturally show repression of Pax7 target genes as we observe in FSHD muscle.

Children are typically unaffected by FSHD, a condition manifesting in late adolescence, with penetrance beginning late in the second decade in males and the third decade in females. Thus it is unsurprising that expected disease mechanisms are not apparent in embryonic stem cell differentiation. Indeed, in the last line of the abstract the authors themselves caution that “**While these studies examine DUX4, PAX3, and PAX7 expression patterns during stem cell myogenesis, they should not be generalized to tissue repair in adult muscle tissue**”. Furthermore, we have recently shown that DUX4 levels increase during regeneration in a mouse model of FSHD (Knopp et al. 2016, *J. Cell Sci.* 129, 3816-31).

Moreover, the authors of the manuscript comment in the discussion:

“We emphasize that there are limitations to this study and that the absence of PAX3/PAX7/DUX4 co-expression in the culture conditions used here does not necessarily mean that DUX4 is never co-expressed with PAX3 or PAX7 at some

point during human development, nor does this study rule out the possibility that DUX4 could compete with PAX3 or PAX7 during regeneration of adult tissues. Our study suggests that conditions that promote expression of PAX3 or PAX7 in myogenic cells do not also promote expression of DUX4 in the same cells. An additional caveat is the possibility that DUX4 expression reduces expression of PAX3 or PAX7 making their expression mutually exclusive but preserving the possibility that they may transiently compete for binding sites [12].”

This article was published after our resubmission to Nature Communications and so we have now included discussion of their observations in the re-revised text.

Reviewer 1 comments: *Minor correction: lines 234 Fig4A should be Fig4A-D, and line 240 Fig4B should be Fig4E.*

RESPONSE: We thank the reviewer for pointing out these errors. These have now been corrected in the re-revised manuscript.

Reviewer #2 (Remarks to the Author):

Reviewer 2 comments: *Minor point: regarding DUX4 repression of MyoD, the authors reference their own paper from 2016. The original demonstration that DUX4 represses MyoD was published in 2008: Bosnakovski et al., EMBO J.*

RESPONSE: We apologise for this oversight and now cite the original demonstration as requested.

REVIEWERS' COMMENTS:

Reviewer #1 (Remarks to the Author):

I am now satisfied with the corrections made by the authors.